# Recurrent chromosome reshuffling and the evolution of neo-sex chromosomes in parrots

Zhen Huang [1], Ivanete De O. Furo[2,3], Jing Liu [4], Valentina Peona [5], Anderson J. B. Gomes [6], Wan Cen[7], Hao Huang[1], Yanding Zhang[1], Duo Chen[7], Ting Xue [7], Qiujin Zhang[1], Zhicao Yue[8], Quanxi Wang[9], Lingyu Yu[10], Youling Chen [1✉], Alexander Suh [5,11], Edivaldo H. C. de Oliveira [12,13,14] & Luohao Xu [4,15✉]

The karyotype of most birds has remained considerably stable during more than 100 million years' evolution, except for some groups, such as parrots. The evolutionary processes and underlying genetic mechanism of chromosomal rearrangements in parrots, however, are poorly understood. Here, using chromosome-level assemblies of four parrot genomes, we uncover frequent chromosome fusions and fissions, with most of them occurring independently among lineages. The increased activities of chromosomal rearrangements in parrots are likely associated with parrot-specific loss of two genes, *ALC1* and *PARP3*, that have known functions in the repair of double-strand breaks and maintenance of genome stability. We further find that the fusion of the ZW sex chromosomes and chromosome 11 has created a pair of neo-sex chromosomes in the ancestor of parrots, and the chromosome 25 has been further added to the sex chromosomes in monk parakeet. Together, the combination of our genomic and cytogenetic analyses characterizes the complex evolutionary history of chromosomal rearrangements and sex chromosomes in parrots.

[1] Fujian Key Laboratory of Developmental and Neural Biology & Southern Center for Biomedical Research, College of Life Sciences, Fujian Normal University, Fuzhou, Fujian, China. [2] Universidade Federal Rural da Amazônia (UFRA) Laboratório de Reprodução Animal (LABRAC), Parauapebas, PA, Brazil. [3] Laboratório de Citogenômica e Mutagênese Ambiental, SAMAM, Instituto Evandro Chagas, Ananindeua, Pará, Brazil. [4] Department of Neurosciences and Developmental Biology, University of Vienna, Vienna, Austria. [5] Department of Organismal Biology, Systematic Biology, Science for Life Laboratories, Uppsala University, Uppsala, Sweden. [6] Instituto Federal do Pará, Abaetetuba, Pará, Brazil. [7] Fujian Key Laboratory of Special Marine Bio-resources Sustainable Utilization, Fuzhou, Fujian, China. [8] Department of Cell Biology and Medical Genetics; International Cancer Center; and Guangdong Key Laboratory for Genome Stability and Disease Prevention, Shenzhen University School of Medicine, Guangdong, China. [9] Fujian Key Laboratory of Traditional Chinese Veterinary Medicine and Animal Health (Fujian Agriculture and Forestry University), Fuzhou, Fujian, China. [10] Annoroad Gene Technology Co., Ltd, Beijing, China. [11] School of Biological Sciences, Organisms and the Environment, University of East Anglia, Norwich, UK. [12] Programa de Pós-graduação em Genética e Biologia Molecular, PPGBM, Universidade Federal do Pará, Belém, Pará, Brazil. [13] Laboratório de Cultura de Tecidos e Citogenética, SAMAM, Instituto Evandro Chagas, Ananindeua, Pará, Brazil. [14] Instituto de Ciências Exatas e Naturais, Universidade Federal do Pará, Belém, Pará, Brazil. [15] Key Laboratory of Freshwater Fish Reproduction and Development (Ministry of Education), Key Laboratory of Aquatic Science of Chongqing, School of Life Sciences, Southwest University, Chongqing, China. ✉email: ylchen@fjnu.edu.cn; luohao.xu@univie.ac.at

The karyotypes and genome sizes of birds have remained considerably stable over more than 100 million years' evolution of modern birds[1–5]. A typical avian karyotype consists of about 40 pairs of chromosomes ($2n = 80$), among which 30 pairs are microchromosomes (smaller than 20 Mb). Among the 12% of bird species with documented karyotype, most have diploid numbers ranging from 76 to 82[1,3,4]. Both cytogenetic mapping and genome assembly have revealed the deep conservation in synteny of both macrochromosomes (larger than 20 Mb) and microchromosomes[6–12]. For instance, emu (a deep-branching bird species) and chicken differ only by one single chromosomal fusion event since their divergence ~100 million years ago[11,13].

Despite the overall conservation, the variation of karyotype is apparent in some bird lineages[3,14]. Many raptor species (birds of prey) have much fewer chromosomes, and particularly, some falcons have a haploid number as low as 20[15–17]. Psittaciformes (parrots, macaws, and alleies) is another bird lineage displaying pronounced karyotype variation[18–20]. For instance, our previous cytogenetic work characterizing the karyotype of monk parakeet (*Myiopsitta monachus*) and blue-fronted amazon (*Amazona aestiva*) revealed their diploid numbers of 48 and 70, respectively[21], suggesting that chromosome rearrangements in birds can occur frequently in less than 30 million years (My)[22]. It is unclear, however, how and why some bird groups have more frequent chromosome rearrangements than others.

It is also unclear how often those chromosomal rearrangements may involve sex chromosomes. The fusion of the sex chromosome and a pair of autosomes can create what is called a 'neo-sex chromosome'. Neo-sex chromosomes have been reported in various organisms, including muntjac deer[23], threespine sticklebacks[24,25], *Drosophila obscura* group species[26], monarch butterflies[27], parasitic nematodes[28], among other animals[29]. Studies across many taxa with neo-XY chromosomes are in line with the suggestion that evolution of neo-Y chromosomes can play a role in resolving sexual antagonism and may affect speciation[25,30,31], though the neo-Y chromosomes often degenerate in the absence of recombination[32]. In birds, the sex chromosome system is female-heterogametic (ZW female, ZZ males) and is highly conserved across bird lineages[33–35]. While the presence of neo-sex chromosomes, to date, has been shown in a few songbirds[36–40] and a cuckoo[41], it remains unclear whether neo-sex chromosomes have evolved in parrots.

In this study, we produced chromosome-level assemblies of monk parakeet and blue-fronted amazon genomes, and re-generated the chromosome-level assembly of budgerigar (*Melopsittacus undulatus*) by re-analyzing the Hi-C data[42]. Our comparative analysis confirms the dynamic evolutionary history of parrot karyotypes which were shaped by frequent and independent inter-chromosomal fusions and fissions. We also discovered and characterized neo-sex chromosomes in parrots, and investigated the evolutionary consequence of the sex chromosome fusions. Finally, we identified one satellite sequence that plays a role in the enlargement of the W chromosome in the monk parakeet.

## Results

### Chromosome-level genome assembly of three parrots

We used chromatin conformation capture (Hi-C) sequencing data to produce chromosome-level assemblies based on a new long-read (PacBio) genome of the monk parakeet and two existing draft genomes of the blue-fronted amazon[43] and the budgerigar[44] (Table 1 and Supplementary Fig. S1). More than 97.5% of contig sequences of monk parakeet were scaffolded into 24 chromosome models (Supplementary Fig. S2), consistent with the known

**Table 1 Genome assemblies of three parrots.**

|  | Monk parakeet | Blue-fronted amazon | Budgerigar |
|---|---|---|---|
| Genome size (Gb) | 1.17 | 1.13 | 1.09 |
| N50 contig (Mb) | 24.5 | 0.4 | 0.05 |
| N50 scaffold (Mb) | 75.7 | 89.0 | 101.4 |
| % assigned to chromosomes | 97.8 | 99.5 | 97.6 |
| BUSCO complete (%) | 94.1 | 93.3 | 93.6 |
| Repeat content (%) | 14.6 | 9.5 | 9.7 |

haploid chromosome number[21]. Among the unanchored contigs (25.8 Mb in length), 89.5% of the sequences are tandem repeats or transposable elements (TEs), suggesting very few non-repetitive genomic sequences are missing from the chromosome assembly. This female genome of monk parakeet also contains a W chromosome that is 13.8 Mb long, harboring 92 protein-coding genes. Given the cytogenetically large size of the W chromosome[21], some heterochromatic sequences are likely missing in the assembly. We identified a candidate centromeric satellite sequence which is 191 bp long and is validated by the fluorescent in situ hybridization (FISH) experiment (Supplementary Fig. S3). For the short-read-based draft genomes of blue-fronted amazon and budgerigar, we anchored 99.5 and 97.6% of the assembled sequences into 29 and 22 chromosome models, respectively. According to the known karyotype of blue-fronted amazon ($2n = 70$)[21] and budgerigar ($2n = 62$)[45], chromosome models of six and nine presumably small microchromosomes, were likely not scaffolded in the respective chromosome assemblies. We further included the kakapo genome assembled by the Vertebrate Genome Project[46] for the following analyses.

To estimate the divergence time among parrot species, we included the genomes of sun parakeet (*Aratinga solstitialis*), kea (*Nestor notabilis*), great tit (*Parus major*), paradise crow (*Lycocorax pyrrhopterus*), chicken (*Gallus gallus*) and emu (*Dromaius novaehollandiae*). The phylogenetic tree built with 1.4 million four-fold degenerate sites is consistent with previous knowledge of parrot phylogeny[47–49], and dated the common ancestor of parrots to about 31.8 My ago (Fig. 1a). The pair of monk parakeet and blue-fronted amazon whose haploid numbers differ by 11 shared a common ancestor 13.5 My ago (Fig. 1a).

### Expansion of a novel TE family in parrots

The proportion of TEs (including LTR and non-LTR retrotransposons) in parrot genomes (mean 9.6%) is slightly higher than that in songbirds (~7.8%, the sister group of parrots) or most other birds[50,51], mainly due to the increased activity of chicken repeat 1 (CR1) non-LTR retrotransposons (Supplementary Fig. S4a). We identified a parrot-specific subfamily of the CR1-E family, named CR1-psi, accounting for about half of the parrot CR1 content (Fig. 1a and Supplementary Fig. S4b, c). While most of the CR1-psi copies were severely 5′ truncated, we detected on average 1860 larger copies of CR1-psi (i.e., >2 kb) that are evolutionarily young and tend to be clustered by species but not by CR1-psi copies, suggesting their recent propagation within species (Fig. 1a and Supplementary Fig. S5).

### Parrot-specific gene loss

To examine the changes in gene content of parrot genomes relative to other birds, we further included a non-avian reptile, the green anole[52], to reconstruct the evolutionary history of bird gene families. While we failed to identify any parrot-specific gene gains, 74 genes were found to be lost in all six parrot genomes but present in the other sampled birds

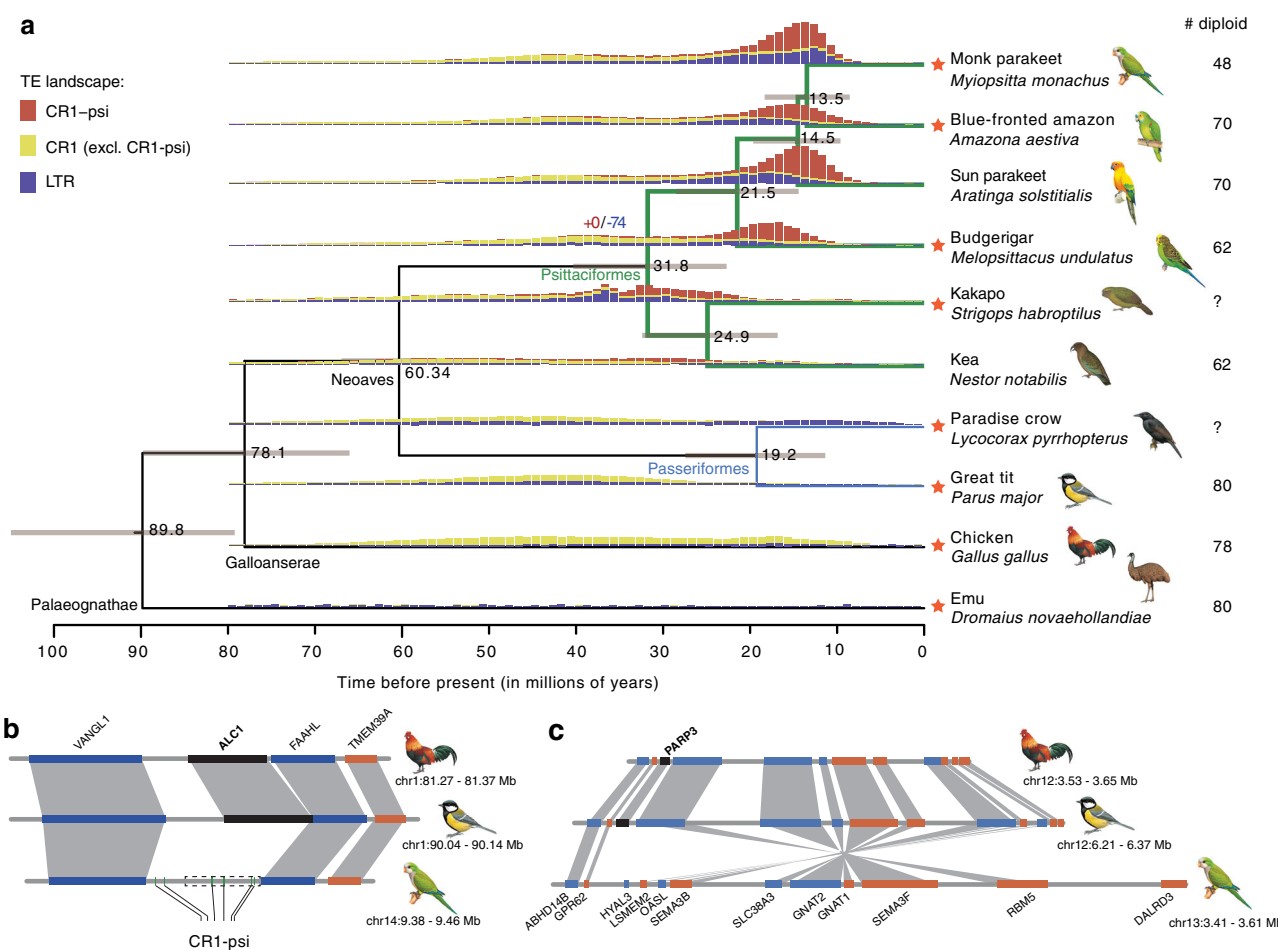

**Fig. 1 Phylogeny and comparative genomics of parrots. a** The phylogeny of nine bird species shows the divergence time (denoted at the nodes) calibrated with fossil records at the node Psittacopasserae (51.81–66.5 My) and Strigopoidea (15.9–66.5 My). The orange asterisks to the left of species names indicate the availability of chromosome-level assemblies. The numbers of gene gains (0, in red) and losses (74, in blue) were denoted at the Psittaciformes node. The vertical bars show the frequency of TE (LTR and CR1) insertions during the evolution of bird species (see details in "Methods"). CR1 is divided into CR1-psi and the other CR1. CR1-psi originated in parrots and continued to propagate in parrot lineages. The numbers of diploid chromosomes are listed in the right panel that shows greater variation in parrots. **b**, **c** The loss of *ALC1* and *PARP3* (highlighted in black) in parrots. Monk parakeet is used to represent the parrot lineage. The synteny of genes around *ALC1* and *PARP3* is illustrated by gray bands connecting orthologs across species. *ALC1* was pseudogenized (indicated by dashed rectangle) due to multiple exon losses (Supplementary Fig. S6). Multiple copies of CR1-psi have been inserted at the *ALC1* locus. Parrots have evolved an inversion around the *PARP3* locus and inversion breakpoint co-localize with *PARP3*. Illustrations reproduced by permission of Lynx Edicions.

(Fig. 1a and Supplementary Table S1). We further confirmed that those 74 genes were not present in the transcriptomes of nine different tissues of monk parakeet (Supplementary Data 1), suggesting they were probably not hidden genes due to incomplete genome assembly[53,54]. Among the lost genes, *ALC1* (Chromo-domain-helicase-DNA-binding protein 1-like, also known as *CHD1L*) and *PARP3* (Poly [ADP-ribose] polymerase 3) are related to repair of DNA damage and maintenance of genome stability[55,56]. *ALC1* is a chromatin remodeler involving in DNA damage response[55] and DNA ends resection[57], and It has been demonstrated that deletion of *ALC1* impacts chromatin relaxation which is a crucial step in response to DNA damage[58]. *PARP3* responses to double-strand breaks[59–61], and has been shown to be involved in the repair of single-strand breaks in avian cells[62]. Studies show that the depletion of *PARP3* delayed the repairs of double-strand breaks[56,63,64] and exacerbated genome instability[65]. Further studies demonstrated that *ALC1* collaborates with *PARP1*, another member of PARP (Poly(ADP-ribose) polymerase) family, on DNA repair[66,67].

To investigate the mechanism of gene loss, we examined the sequences harboring the homologous genes in non-parrot genomes and compared their synteny with parrot genomes to detect genomics changes at the loci of parrot gene losses. Among the 74 lost genes, 24 reside in conserved synteny blocks (collinear gene order) across bird species (Fig. 1b and Supplementary Fig. S6), and the gene loss of five of them, including *ALC1*, coinciding with CR1-psi insertions. At the locus where *ALC1*, for instance, has been pseudogenized through multiple independent exon deletions (Supplementary Fig. S7), we detected multiple copies of CR1-psi (Fig. 1b). This likely reflects the nature of a mutational hotspot in this region that may have initially led to pseudogenization of *ALC1*. Alternatively, it is also possible that sequence deletion due to non-allelic homologous recombination between CR1-psi copies contributed to exon losses[68]. For six of the 74 lost genes, we found that their homologs in non-parrots are located at the breakpoints of parrot-specific rearrangements. (Supplementary Data 2). For instance, *PARP3* is located at the boundary of a parrot-specific inversion involving 9 genes (Fig. 1c).

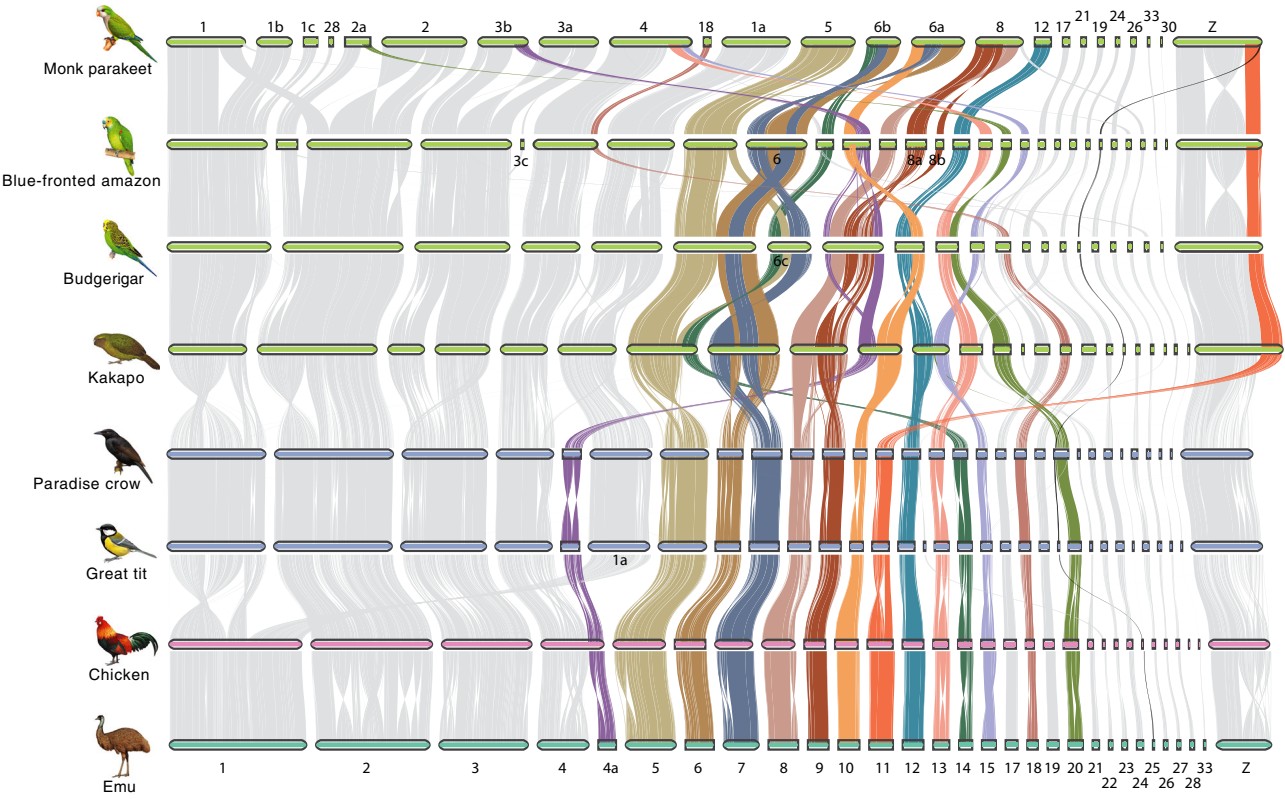

**Fig. 2 Frequent chromosomal rearrangements across parrot genomes.** Pairwise whole-genome alignments across eight chromosome-level assemblies of bird genomes. Each horizontal bar represents one chromosome, and chromosome IDs of chicken were labeled at the bottom; for parrot genomes some chromosomes were renamed following chromosome fissions. Chromosomes that experienced recurrent chromosomal rearrangements are highlighted in colors. Chromosome 11 and 25 are highlighted in color which were fused to the sex chromosome in monk parakeet. Illustrations reproduced by permission of Lynx Edicions.

This suggests the potential role of chromosomal rearrangements (inversions) in gene loss[69,70].

**Frequent and independent inter-chromosomal rearrangements.** The decreased number of chromosomes in parrots compared to the typical number in avian karyotypes suggests the occurrence of multiple chromosomal fusion events in this lineage. To reconstruct the evolutionary history of chromosomal changes, we aligned the chromosome-level assemblies of four parrots and four outgroups that have chromosome-level assemblies, namely two passerines (great tit and paradise crow, representing Neoaves), chicken (Galloanserae), and emu (Palaeognathae), thus covering all three major bird clades. Between the four non-parrots, typically no more than two events of inter-chromosomal rearrangements can be found in pairwise comparisons, but within the parrot lineage multiple chromosomal fusions or fissions events occurred independently (Fig. 2). We found that only three chromosomal fusions (chr6+chr7, chr8+chr9, and chr11+chrZ) and zero fission are shared by all parrots (Fig. 2 and Supplementary Fig. S8). In addition, none of the chromosomal changes are common to the ancestral lineage leading to monk parakeet and blue-fronted amazon except for a fission of chr2 (Fig. 2 and Supplementary Fig. S8). In other words, most (32 out of 38) chromosomal rearrangements are specific to the lineages leading to each sampled species and occurred in less than 31.8 My. Some chromosomal segments have experienced frequent rearrangements in one lineage. For instance, an ~18 Mb sequence (chr2A) was split from the ancestral chr2 and further experienced one fusion and one fission in the lineage leading to monk parakeet (Fig. 2). Monk parakeet and budgerigar have lower haploid

chromosome numbers than blue-fronted amazon (24 and 31 vs. 35), which can be explained by more chromosomal fusions (Fig. 2). In particular, we identified eight events in which a microchromosome has fused to a macrochromosome, but no inter-microchromosome fusion events have been detected, in agreement with the suggestion by studies in falcons that reduction of chromosome number is mainly due to fusions of microchromosomes to macrochromosomes[71]. In addition, chromosomal fissions likely led to two and three new microchromosomes in monk parakeet and blue-fronted amazon, respectively (Fig. 2), but such newly formed microchromosomes seem to be rare in budgerigar or kakapo or other birds.

The frequency of chromosomal rearrangements varies among chromosomes, with some chromosomes experiencing repeated and independent rearrangements. We found that 12 chromosomes (chr4A, chr5, chr6, chr7, chr8, chr9, chr10, chr13, chr14, chr15, chr18, and chr20) have experienced at least two independent chromosomal changes in parrots. For example, chr10 was fused with chr12 in budgerigar, in blue-fronted amazon and kakapo with chr4a and in monk parakeet with chr6a (Fig. 2). Those more frequently rearranged chromosomes tend to have an intermediate chromosome size (20–40 Mb) (Fig. 2).

**Breakpoints of inter-chromosomal rearrangements.** We next asked whether the chromosomal fusions were adaptive[72,73], and whether fissions occurred at random positions. To evaluate whether the newly joined chromatins have established frequent interactions due to new *cis*-regulation across the fusion sites, we examined the fusion sites for insulation scores (ISs). If one locus has a higher IS, it is less likely located at the boundaries of

topologically associating domains (TADs). At the fusion sites, we found the ISs tend to be lower, though only significantly so in blue-fronted amazon (Wilcoxon test, $p = 0.015$) (Supplementary Fig. S9a), suggesting interactions across the fusion sites are still infrequent. Similarly, the breakpoint of fissions tend to be located in regions with lower ISs in the pre-fission chromosomes (Wilcoxon test, $p < 0.05$) (Supplementary Fig. S9b), suggesting of strong interacting barriers between the two flanking regions. For instance, a fission breakpoint at 107 Mb on chr1 of blue-fronted amazon has the lowest IS along the chromosome, separating two mega-scale flanking chromatin domains (Supplementary Fig. S9c). This suggests that the location of the fission breakpoint is not random, but at the boundary of TADs, thus such fission, as occurred in monk parakeet (Supplementary Fig. S9c), probably has a limited impact in disrupting *cis*-regulation. Together, our analyses suggest that chromosomal rearrangements in parrots were likely not driven by positive selection, but shaped by purifying selection.

**Chromosomal fusions led to the formation of neo-sex chromosomes.** The Hi-C-based chromosome-level assemblies and whole-genome alignments (Fig. 3a) reveal that chr11 was fused to the ZW sex chromosomes in the ancestor of parrots, and one additional fusion of chr25 to the sex chromosomes found only in monk parakeet. The FISH experiments using the probes of chicken chr11 and chr25 sequences further validated the chromosomal fusions involving sex chromosomes (Fig. 3b and

Supplementary Fig. S10). It was previously observed that chr25 fused to a large macrochromosome[20], but the latter was wrongly identified as chr4 instead of chrZ. Unlike the scenario of neo-sex chromosome formation in songbirds that involved translocating a part of macrochromosomes[36–40], both chr11 and chr25 are microchromosomes, and their entire lengths (23.7 and 2.8 Mb respectively) were added to the sex chromosomes (Fig. 3a). To assess whether and to what extent the neo-sex chromosomes are differentiated, we mapped the female sequencing reads to Z chromosomes, with the anticipation that fully differentiated sex chromosomes with divergent Z and W sequences would show reduced coverage. We found the chr11-derived neo-sex chromosome to be hemizygous and fully differentiated but not the chr25-derived one in monk parakeet, consistent with the more recent fusion of chr25. In fact, for the chr25-derived neo-sex chromosome we were not able to assemble the Z- and W-linked sequences separately, likely due to little sequence divergence between them (therefore the coverage is autosome-like, Fig. 3c), though it is possible chr25 only fused to the Z chromosomes. Resequencing data from five additional parrot species: Carolina parakeet (*Conuropsis carolinensis*), Hispaniolan amazon (*Amazona ventralis*), Puerto Rican amazon (*Amazona vittata*), red-and-green macaw (*Ara chloropterus*) and cockatiel (*Nymphicus hollandicus*) (Supplementary Fig. S11) further support that the fusion of chr11 to sex chromosomes is shared by all parrots.

The female coverage patterns also indicate that the old ZW sex chromosomes are fully differentiated, with only a small

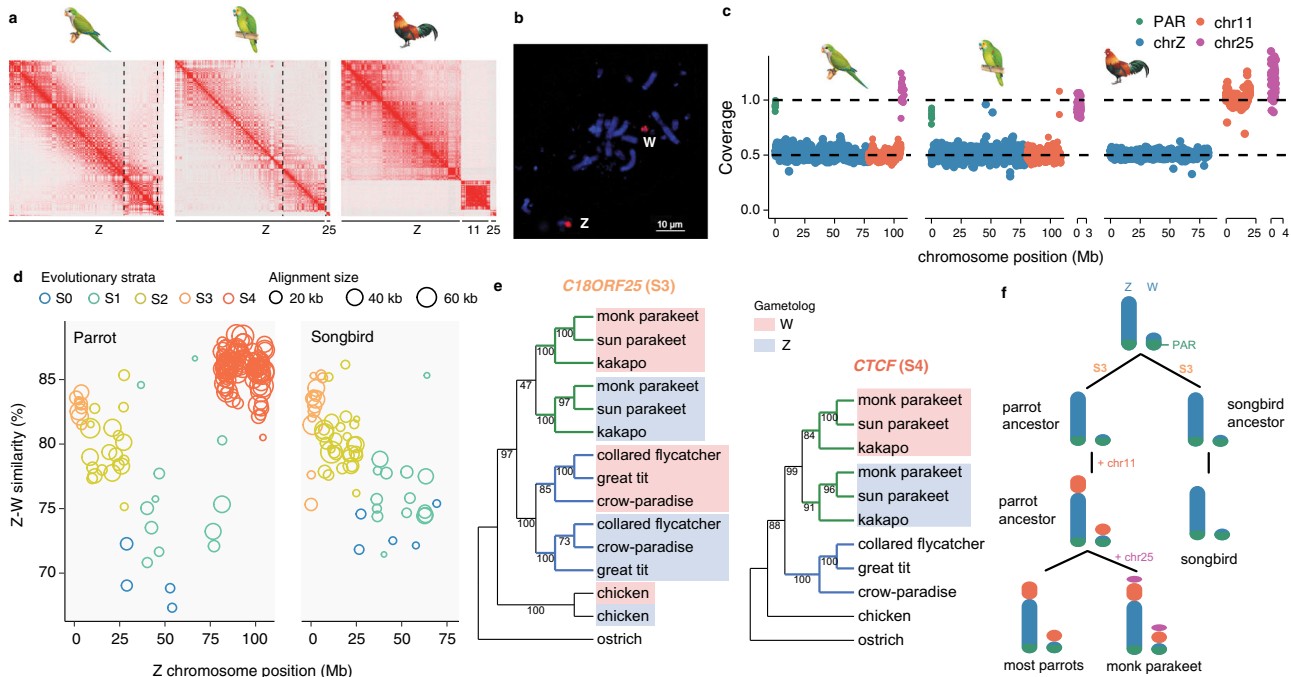

**Fig. 3 The Evolutionary history of parrot sex chromosomes. a** The Hi-C contact map of chromosomes that are homologous to chicken chr11, chr25, and chrZ. The three chromosomes are presented in three chromosome territories in chicken, but only one in monk parakeet and two in blue-fronted amazon. **b** The FISH images for the probes of chicken chr11 hybridization in monk parakeet. The FISH result for chicken chr25 probe is shown in the Supplementary Fig. S10. **c** The female sequencing coverage for the chromosomes homologous to chicken chr11, chr25, and chrZ. In monk parakeet the blue-fronted amazon the coverage of the part of chrZ homologous to chicken chr11 is reduced by half, compared to PARs or autosomes. **d** Sequence divergence of the Z and W chromosome reveals the pattern of evolutionary strata. Each dot represents a 100-kb sliding window along the Z chromosome. The S0 and S1 were defined based on the homology of Neoaves S0 and S1 (Xu, Auer et al. 2019). Songbirds (right panel) have four evolutionary strata and parrots (left) have a similar stratum pattern but have one additional stratum (S4) due to the fusion of chr11. **e** Phylogeny of the Z-W gametologs for a S3 (left panel) and a S4 (right) gametologous gene. Parrot Z- and W-linked gametologs are clustered together respectively, and share a common ancestor, suggesting common origins of S3 and S4 in parrots. Additional gene trees are given in the Supplementary Fig. S13. **f** A schematic diagram depicting the evolutionary history of sex chromosomes since the common ancestor of songbirds and parrots. Whether chr25 has been added to the chrW remains to be verified. Illustrations reproduced by permission of Lynx Edicions.

pseudoautosomal region (PAR) of ~500 kb still recombining between the Z and W chromosomes (Fig. 3c). It is known that the sex chromosomes diverge from each other following the arrest of recombination that often occurs in a step-wise manner, forming the so-called evolutionary strata[34,74]. It was previously demonstrated that all Neoaves (including parrots and songbirds) shared the first three evolutionary strata (S0–S2), with the fourth stratum (S3) often occurring independently among Neoaves lineages[34]. To demarcate the boundary of S3, we closely examined divergence levels (measured by sequence similarity) between the old Z and W chromosomes, and found that the parrot S3 boundary is shared with songbirds[50] (Fig. 3d and Supplementary Fig. S12). Phylogenetic analysis of the homologous Z-W gene pairs (gametologs), however, suggests that S3 evolved independently in songbirds and parrots (Fig. 3e).

The chr11-derived neo-sex chromosome formed a single stratum (S4), exhibiting Z-W sequencing similarity of ~86%, higher than those of other evolutionary strata (Fig. 3d). This suggests that the chr11 was added when S3 was already formed, i.e., the old chromosomes were differentiated (Fig. 3f). It needs to be noted that chr11 (and chr25) were added to the ZW chromosomes at the differentiated end, i.e., not at the PAR (Fig. 3f), therefore having no impact on the existing PAR. This newly formed S4 retained 16.9% of its original gene content on the neo-W chromosome, compared to only 2.6% on the old W chromosome (Supplementary Table S2). The phylogenetic analysis of gametologs from major parrot lineages confirms the origin of the chr11-derived S4 at the ancestor of parrots (Fig. 3e and Supplementary Fig. S13).

**Non-adaptive evolution of the neo-sex chromosome**. Next, we asked whether the newly acquired female-specific neo-W chromosome (S4) has evolved a gene repertoire beneficial to females and acquired female-specific or female-biased expression, similar to what had been reported for the neo-Y chromosome of *D. miranda*[30]. The surviving genes ($n = 65$) on the W-linked S4 all have Z-linked origins, except for a retrogene (*DYNLRB1*) derived from chr12 through LTR-mediated retroposition (Supplementary Fig. S14a)[75]. This intronless retrogene, however, does not seem to be expressed from the W (Supplementary Fig. S14b). The expression profile of the S4 Z-W gene-pairs across eight different tissues appears to be highly correlated (Fig. 4a, b), suggesting little alteration in gene expression of the W-linked gametologs since their arrival on the W chromosome. However, we detected global downregulation of the W-linked gametologs relative to the Z-linked counterpart (Fig. 4a, b), likely due to the more inactive chromatin environment of the W chromosomes[11,33]. Despite that, we identified three genes that are upregulated across tissues, but only one of them belongs to S4 (Fig. 4b).

The loss of W-linked genes also leads to imbalanced dosage of the proto-sex-linked genes relative to autosomes. The avian sex chromosomes are known to have not evolved global dosage compensation and balance[76,77]. Consistently, our results suggest that S4 has not evolved a mechanism to fully compensate for the gene dosage across tissues, with male-to-female expression ratios ranging from 1.62 to 2.11 (Fig. 4c and Supplementary Table S3).

Accompanied with rapid gene loss and downregulation, the W-linked S4 has already increased the TE content from 12.8 to 50.1% (Fig. 4d). The accumulation is mainly driven by LTR element insertions which can be quite long (>5 kb) and tend to remain in their full-length form likely because of the reduced recombination and low efficacy of selection[78,79]. The relatively young age of S4 (formed ~31.8 MY ago) also provides a unique window into the temporal dynamics of TE accumulation on the non-recombining sex chromosome, and we demonstrate that

LTRs rapidly accumulate on the younger strata while CR1 accumulates more slowly, but over time CR1 gradually increases its proportion on older strata (Fig. 4d).

**Enlargement of the W chromosome associated with expansions of satellite DNA**. The size of the W chromosome in monk parakeet is unusually large and is similar to that of the Z (Supplementary Fig. S15a)[21] while in most parrots the W chromosomes are much smaller[80]. Since the chr11 and likely chr25 were added to both the Z and W chromosomes in monk parakeet, the enlargement of the W chromosome is not due to fusions alone, but likely expansions of repeat sequences[21]. To identify W-specific repeats, we compared the k-mers from female and male sequencing reads, and found that the top 4 most frequent k-mers in females were absent in males (Supplementary Fig. S15b). Those k-mers were derived from a 20-bp satellite (SatW20) which was estimated to have 194,438 copies in the genome, presumably all on the W chromosome. To validate the specificity of SatW20 to the W chromosome, we performed FISH experiments using the monomer of SatW20 as a probe, and confirmed its W-specific binding (Supplementary Fig. S15a).

To unravel the origin of SatW20, we analyzed the composition of repeats of chr11 and chr25 in three other bird genomes (representing three major bird clades) assembled with long-reads. In all these birds, chr25 but not chr11 has a considerably large portion of satellite DNA, compared with other chromosomes (Supplementary Fig. S15c). Coincidentally, chr25 was added to monk parakeet specifically. The satellite families of chr25, however, differ among species, indicating rapid turnover of satellite repeats. We were unable to find SatW20 sequences in any of the bird genomes except for the W chromosome of monk parakeet. This suggests that SatW20 may have a recent origin following the fusion of chr25 into the ZW chromosome in monk parakeet.

## Discussion

Through comparative analyses of chromosome-level genomes of four parrots, combined with cytogenetic analyses, we uncovered numerous events of chromosomal rearrangements in each of the parrot genomes. Some microchromosomes were missing in the genome assemblies of blue-fronted amazon and budgerigar, so we may have underestimated the frequency of chromosomal changes if the microchromosomes also experienced any rearrangements. The majority of the chromosomal rearrangements are lineage-specific, suggesting that karyotypic changes have been regularly taking place in the course of parrot diversification. Apart from the frequent chromosomal changes, the parrot genomes have a slightly higher proportion of TEs than songbirds (9.6% vs. 7.8%) due to recent CR1 proliferation, and we speculate that recent CR1-psi accumulation may have provided substrates for ectopic recombination events leading to the loss of at least some of the 74 conserved genes. Among the lost genes, *ALC1* and *PARP3* are known for their roles in genome stability and DNA double-strand repair[55–62,65,66,81]. This led to our hypothesis that the frequent chromosomal rearrangements may represent one evolutionary consequence of the loss of both *ALC1* and *PARP3*, which we speculate to have happened through deletions associated with CR1-psi and inversions, respectively. If this were the case, chromosomal rearrangements are perhaps not fixed by natural selection, but rather through genetic drift. It remains elusive whether TEs play a causal role in promoting gene loss or chromosomal rearrangements, and we cannot rule out the possibility that TE expansion simply correlates with other evolutionary events that are specific to parrots.

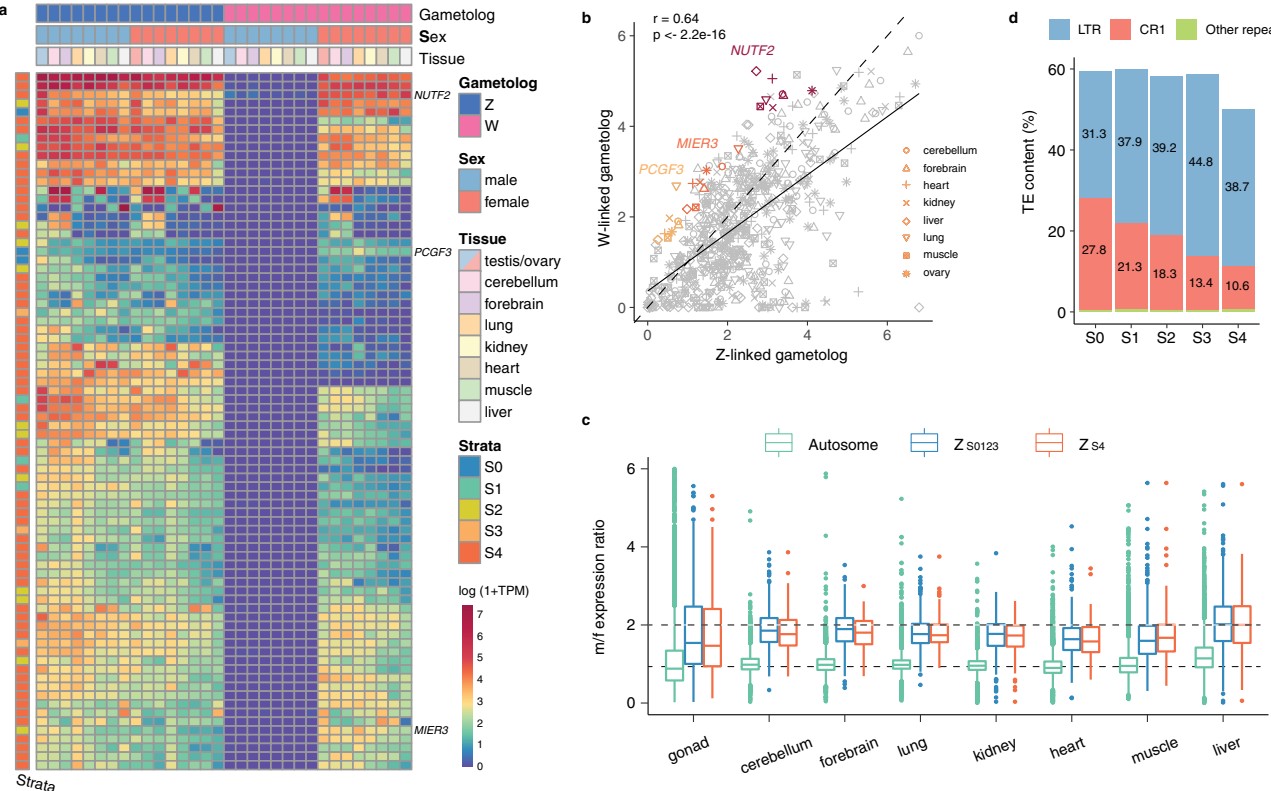

**Fig. 4 Non-adaptive evolution of the neo-sex chromosome. a** A heatmap showing the expression (measured by log (1+TPM)) of Z-W gene pairs across male and female tissues in monk parakeet. Each row represents one gene. The top three panels annotate whether a gene is Z- or W-linked and in which tissue/sex it is expressed. The left panel annotates which stratum a gene belongs to. The W-linked gametologs are present only in females (purple in male), but their expression profiles are similar to their Z-linked homologs. **b** The X- and Y-axis show the expression of Z- and W-linked gametologs, respectively. The solid line represents the regression relation between the expression of the Z and W, while the dashed line indicates the situation of equal Z-W expression. **c** Male/female expression for Z-linked genes on the both old ($n = 529$) and new ($n = 122$) sex chromosomes, as well as autosomal genes ($n = 7,562$). The boxplots show the first quartile, median, and third quartile values. **d** Temporal dynamics of TE content on the monk parakeet W chromosome divided into five evolutionary strata. LTRs are more abundant in younger strata.

One specific form of inter-chromosomal rearrangements is the fusion of sex chromosomes with autosomes, which has occurred one time at the parrot ancestor and one additional time in monk parakeet. We previously demonstrated that female-specific selection has limited influence on the bird sex-specific W chromosomes whose gene content is primarily shaped by purifying selection[33]. Consistently, we did not detect signals of female-specific selection favoring the addition of autosomal female-beneficial genes into the female-specific W chromosomes. In fact, the neo-W chromosomes degenerate and rapidly accumulate TEs, with primarily housekeeping genes surviving, behaving like the old W chromosome[33,78,82]. The rapid gene losses further impose the challenge of dosage imbalance for female genomes which lack mechanisms of global dosage compensation[76,77]. Moreover, the addition of chr25 that is repeat-rich in birds into the sex chromosome in monk parakeet has probably contributed to the runaway expansion of satellite DNA and heterochromatin on the W chromosome which likely has a deleterious effect for the female genomes, e.g., genome instability[83,84].

## Methods

**Sample collections**. In this study, we collected one female monk parakeet for long-read and Hi-C sequencing, and additional three males and females for RNA-seq. The RNA-seq data was used for genome annotations as well as quantifying the expression of sex-linked genes. We also collected blood samples from one female blue-fronted amazon for Hi-C sequencing, and one female red-and-green macaw (*Ara chloropterus*) and one female cockatiel (*Nymphicus hollandicus*) for low-coverage resequencing. The full sample information can be found in the Supplementary Data 1. All animals were collected from Fuzhou Olsen Agriculture

CO.LTD. The Institutional Animal Care and Use Committee (IACUC) of Fujian Normal University has approved the animal ethics.

**Monk parakeet long-read sequencing**. Parallel genomic DNA extractions were performed on blood from a single female monk parakeet individual, using the DNeasy Blood & Tissue Kit (QIAGEN, Valencia, CA) following the manufacturer's instructions. The DNA was quantified using the Qubit 2.0 Fluorometer (Thermo Fisher Scientific, Waltham, MA) with its standard protocol. To check for molecular integrity, each DNA was run on the 2200 TapeStation (Agilent Technologies, Santa Clara, CA) following the manufacturer's protocol. The extracted DNA was used to construct a 20 kb PacBio SMRTbell™ library prepared with the Sequel Sequencing Kit 3.0, according to the released protocol from the PacBio Company. The library was sequenced on the Sequel II machine by Annoroad Gene Technology company (Beijing, China), producing ~80 Gb reads in one SMRT cell.

**Genome assembly**. We used Falcon (pb-assembly v0.0.4)[85] to assemble the long reads into contigs. Reads shorter than 8000 bp were discarded. The full configuration has been deposited to Github (see Code Availability). The haplotigs were removed using the program purge_haplotigs (v1.0.4)[86] with the parameter '-a 60'. To polish the assembly with long reads, we mapped the reads to the draft assembly with pbmm2 (0.12.0) which used minimap2 (2.15-r905)[87] for alignment, and then used the arrow algorism for polishing. The mapping-and-polishing process was repeated twice.

**Monk parakeet short-read sequencing**. We produced ~30X Illumina short-read data from the same female individual in order to further polish the assembled contigs. DNA from the muscles were extracted. Then, a paired-end library with an insert size of 350 bp was constructed and sequenced by Annoroad Gene Technology company (Beijing, China) in accordance with the manufacturer's protocol using the Illumina HiSeq X Ten platform. We mapped the raw reads to the contigs using BWA-MEM (0.7.16a-r1181) with default parameters. Then pilon (1.22)[88]

was used to fix the bases and indels with '--minmq 30 --mindepth 20 --diploid' options. The short-read polishing process was also repeated twice.

**Hi-C scaffolding**. The muscle tissues of monk parakeet were fixed using formaldehyde for 15 min at a concentration of 1%. The chromatin was cross-linked and digested using the restriction enzyme *MobI*, then blunt end-repaired, and tagged with biotin. The DNA was religated with the T4 DNA ligation enzyme. After ligation, formaldehyde crosslinks were reversed and the DNA purified from proteins. Biotin-containing DNA fragments were captured and used for the construction of the Hi-C library. Finally, 350 bp paired-end libraries constructed from DNA were sequenced on the Illumina HiSeq X Ten platform, producing ~80 Gb of sequencing data (Supplementary Data 1). We used the 3D-DNA pipeline (180114)[89] to join the contigs into chromosomes. First, we used the juicer (1.7.6) pipeline[90] to map the Hi-C raw reads against the contigs. Then, we ran the run-asm-pipeline program to do the scaffolding with the options --editor-coarse-resolution 500000 --editor-coarse-region 1000000 --editor-saturation-centile 2 -r 1. The Hi-C contact map based on the draft chromosomal assembly was then visualized in Juicebox which also allowed for manual adjustment of the orientations and order of contigs along the chromosomes. During this process, some misplaced sequences (due to Falcon assembly errors) were cut off from the original contigs and were re-joined to the correct ones.

**Chromosomal assembly of blue-fronted amazon and budgerigar**. We used the same protocol of Hi-C library preparation and followed the same procedure of Hi-C sequencing for blue-fronted amazon as in monk parakeet. For budgerigar, we downloaded the Hi-C data from Cooke et al. (2017)[42] to do scaffolding based on the draft genome[44]. We then used the same pipeline of Hi-C scaffolding as we did for monk parakeet (Supplementary Fig. S1).

**Monk parakeet genome annotation**. To annotate the repeat content, we first used RepeatModeler2 (2.0)[91] to predict and classify TEs throughout the genome. Tandem repeats were predicted with Tandem Repeats Finder (4.09)[92] and the raw results were filtered by the pyTanFinder pipeline[93]. The newly predicted families of TEs and tandem repeats were then combined with recently curated bird repeat library (including multiple passerines, kakapo, hummingbird, chicken, emu)[10] to annotate repeats using RepeatMasker (4.0.7). The repeat-masked genome was then used for the gene model annotation with the MAKER pipeline. The protein sequences from budgerigar, hooded crow and chicken were downloaded from NCBI Refseq database. We mapped the raw RNA-seq reads from nine different tissues of monk parakeets using HISAT2 (2.1.0)[94], and assembled the transcriptomes using StringTie (1.3.3b) with options '-m 300 -a 12 -j 5 -c 10'. The gene models were predicted by MAKER based on the alignments of protein sequence and Stingtie transcripts against the genome. To further polish the gene models, we assembled the de novo transcriptome using Trinity (2.8.4)[95] and modified the gene models using the PASA pipeline (v2.4.1)[96] with --stringent_alignment_overlap 30 --gene_overlap 50.

**Phylogenomics**. We used Last (v1170)[97] to align the genomes of eight birds (Supplementary Table S4), including blue-fronted amazon, budgerigar, sun parakeet[47], kea[1], kakapo[46], chicken[98], great tit[99], paradise crow[10], against that of the monk parakeet. The multiple alignment results were then merged with Multiz (v11.2)[100]. A total of 1.4 million four-fold degenerate sites were extracted to construct the phylogenetic tree, using IQ-TREE (2.0-rc1) with 1000 bootstrap replicates. We then used BASEML (4.9j)[101] to estimate the overall mutation rate with the time calibration on the root node (102 MY)[49]. General reversible substitution model and discrete gamma rates were estimated by the maximum likelihood approach under strict clock. The divergence time was then estimated using MCMCtree (4.9j)[101], with two soft-bound calibration time points: 15.9–66.5 My for the ancestral node of kea and kakapo and 56–64.5 My for the Passeriformes-Psittaciformes split[49].

**CR1-psi**. The draft consensus sequence of CR1-psi, a parrot-specific CR1 element named in this study, was predicted by RepeatModeler2. To curate the consensus sequence, we searched and extracted the homologous sequences in the monk parakeet genome, with an extension of 100 bp to both downstream and upstream directions. The extracted sequences were aligned using MAFFT (7.397)[102] and the alignment was visualized with Aliview (1.25)[103]. We manually inspected the alignments and decided the boundaries of the consensus. The consensus sequence is available in the Supplementary Table S5. We used the parseRM program[4] to estimate the timing of TE insertions at the family level. To classify the CR1-psi, the phylogeny of known CR1 elements[104,105] was constructed using FastTree (2.1.11)[106] with default parameters.

**Chromosomal rearrangement**. We used the MUMmer (4.0.0.beta2)[107] tool nucmer to perform the pairwise whole genome alignment (see Fig. 2 for the alignment pairs) with the parameter "-b 400". The alignments were filtered to keep the one-to-one best hits using delta-filt from the MUMmer package. Unanchored scaffolds

were excluded from the alignments. The alignments were formatted so as to be used by the MCscan pipeline[108] for synteny visualization.

**FISH experiment**. Chromosome preparations were obtained from fibroblasts from feather pulp biopsies of a male and a female of monk parakeet, using standard cell culture and colcemid/hypotonic solution treatment protocols[21]. Slides were prepared using the air-drying method. FISH with repetitive sequences followed[109], and used sequences directly labeled with CY3 as probes. Probes from chicken (GGA) were used to detect segments homologous to chr11 and chr25. The probes of SatW20 and the 191 bp centromeric sequence were synthesized at Exxtend Biotecnologia Ltda (Paulinia, SP, Brazil). We performed FISH experiments using a whole chromosome paint corresponding to chr11, obtained by flow cytometry and labeled with CY3 by DOP-PCR, and BACs (Bacterial Artificial Chromosomes) containing fragments of chr25, labeled with CY3 (short arm) and FITC (long arm), as described previously[6,21]. Slides were analyzed with a Zeiss Imager Z2 epifluorescence microscope equipped with a cooled CCD camera and appropriate filters. Images were captured using Axiovision 4.3 (Zeiss), and edited with Photoshop (21.0.0).

**Insulation scores**. We calculated insulations scores in order to investigate whether evolutionary breakpoints tend to locate at TAD boundaries. To do so, first we mapped Hi-C reads with the Juicer pipeline. We generated the matrix of Hi-C interaction using the 'dump' command of Juicebox at 100-kb resolution with Knight-Ruiz (KR) normalization. We required the interaction count of at least 10 between the pairs of 100 kb interacting windows. This filtered Hi-C matrix was used to visualize the Hi-C contact map. To calculate the insulation scores of TAD boundaries, we used BWA-MEM (-A 1 -B 4 -E 50 -L 0 -t) to map each of the Hi-C read-pairs to the genomes, and generated the Hi-C matrix using the hicBuildMatrix command of HiCExplorer (2.2.1.1). Using the hicFindTADs command, we identified the TAD boundaries and calculated the insulation scores at 200-kb resolution.

**Evolutionary strata**. The Z and W chromosome sequences were masked for repeats prior to LASTZ (1.04)[110] alignment. We used a relaxed parameters (--step=19 --hspthresh=2200 --inner=2000 --ydrop=3400 --gapped-thresh=10000) to align the W chromosome to the Z, and alignment chains and nets were further produced to join the syntenic fragmented alignments in longer alignments. Then we filtered out the alignments that are too short (less than 65 bp) or have too low sequence similarity (less than 60%) to reduce the false positive rate. Alignments with unusually high sequence similarity (larger than 96%) were also removed because they might be derived from unmasked simple repeats. We then calculated the sequence similarity between the Z and W over 100-kb sliding windows along the Z chromosome. Because the first (S0) and second (S1) strata are shared by all Neoaves, we used the previously identified sequences of songbird S0 and S1[50] to demarcate the boundaries of S0 and S1 in parrots. We closely examined the Z-W sequence similarity other than the S0/S1 regions, and identified the putative stratum boundary of S2 and S3 at 4.1 Mb. We then used the phylogeny of the Z-W gametologs in the left and right of the putative boundary to test whether they belong to different strata. We used MAFFT (7.427)[102] to align the coding sequencing of Z-W gametologs and the homologous genes of chicken, ostrich, great tit, collared flycatcher, and paradise crow, with default parameters. We used IQ-TREE (2.0-rc1)[111] to perform the phylogenetic analysis, with the substitution model automatically selected. Bootstrapping was repeated 100 times. For great tit and budgerigar, since the genomes were derived from male individuals, we downloaded the female RNA-seq data (SRR2170826 [https://www.ncbi.nlm.nih.gov/sra/?term=SRR2170826] and SRR5336544 [https://www.ncbi.nlm.nih.gov/sra/?term=SRR5336544][42] respectively) from NCBI SRA to assemble the transcriptomes using Trinity (2.8.4)[112], in order to assemble the sequences of W-linked genes. The following options were used in Trinity assembly: "--path_reinforcement_distance 30 --trimmomatic". We then mapped the RNA-seq reads back to the transcripts with HISAT2 (the above mentioned parameters were used), and the alignments were closely examined in IGV (2.4.3)[113] for potential assembly errors.

**Re-sequencing**. DNA from the feathers of red-and-green macaw and cockatiel were extracted using EasyPure® Genomic DNA Kit (Transgen Biotech, Beijing, China). The re-sequencing data was generated in the same way as described in the section "Monk parakeet short-read sequencing". Raw read data from Carolina parakeet[47], Hispaniolan amazon[114], Puerto Rican amazon[114] were downloaded from NCBI. We did not trim these reads since they were used only to calculate sequencing coverage.

**Sex chromosome gene expression**. We collected nine tissues from three males and three females of monk parakeet respectively (Supplementary Data 1). The extracted RNA was quantified using the 2100 Agilent Bioanalyzer (Agilent) before library construction. A total quantity of 3 μg RNA per sample was used for paired-end library construction. Sequencing libraries were generated using NEBNext®Ultra™ RNA Library Prep Kit for Illumina® (#E7530L, NEB, USA), following the manufacturer's recommendations. Briefly, mRNA was purified from total RNA using poly-T oligo-attached magnetic beads (Beckman Coulter). Fragmentation

was carried out using divalent cations under elevated temperatures in NEBNext First Strand Synthesis Reaction Buffer (5X). First-strand cDNA was synthesized using random hexamer primer and RNase H. Second-strand cDNA synthesis was subsequently performed using buffer, dNTPs, DNA Polymerase I, and RNase H. The library fragments were purified with QIAQuick PCR kits (Qiagen) and eluted with Elution buffer. After terminal repair, poly(A) sequence and adapter were implemented. The resulting cDNA libraries were run on a 2% agarose gel and bands of approximately 250 bp were excised and used for paired-end sequencing on an Illumina HiSeq X ten platform by Annoroad Gene Technology Co. Ltd. Trimming for low-quality bases and adapters was also completed by Annoroad Gene Technology Co. Ltd according to the filtering pipeline described in[115].

We use HISAT2 (2.0.4) to map the RNA-seq reads against the monk parakeet genomes with the options -k 4 --max-intronlen 100000 --min-intronlen 30. After sorting the alignments, we used featureCounts (v1.5.2) to count the reads mapped to the annotated gene models. Then TPM (transcripts per million) were calculated to normalize the expression levels for each tissue. The mean TPM values were calculated over the biological replicates.

**Reporting summary**. Further information on research design is available in the Nature Research Reporting Summary linked to this article.

## Data availability

The genome assemblies and sequencing data are deposited at NCBI under the accession PRJNA679636. A full list of accession IDs is available in the Supplementary Data 1.

## Code availability

The custom scripts used in this study have been deposited at Github (https://github.com/lurebgi/monkParakeet)[116].

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

## Acknowledgements

We thank Zongqin Xu (Tianshen Bird Import and Export Trading Co., Ltd.) for providing the feathers of red-and-green macaw and cockatiel, and thank Qi Zhou and Simone Fouché for comments. We also thank Carla Canedo Ribeiro and Darren Griffin for testing FISH probes. Z.H. is supported by the scientific research innovation program "Xiyuanjiang River Scholarship" from the College of Life Sciences, Fujian Normal University. L.X. is supported by the Erwin Schrödinger Fellowship (J4477-B) from the Austrian Science Fund (FWF). Computational resources were provided by the Life Science Compute Cluster (LiSC) from the University of Vienna.

## Author contributions

L.X., H.Z., and E.O. conceived this study. L.X., Z.H., W.C., H.H., J.L., V.P., and A.S. performed the data analyses; I.F., A.G., and E.O. contributed to the cytogenetic works. Y.Z., D.C., T.X., Q.Z., Z.Y., Q.W., and L.Y. assisted data collections. L.X. and Y.C. supervised this study. L.X. wrote the manuscript draft that has been revised by Z.H., V.P., A.S., and E.O.

## Competing interests

The authors declare no competing interests.
