## [Peer Review File · Nature Communications]

Recurrent chromosome reshuffling and the evolution of neo-sex chromosomes in parrotsReviewers' Comments:

Reviewer #1:

Remarks to the Author:

This paper offers a potentially fascinating findings about the frequency and extent of chromosomal changes in parrots. I found some of the statements about how how specific some of the chromosomal changes are to one or other lineage gave me pause. I wondered whether more extensive clade sampling of the parrot phylogeny would dictate modifications to those statements. Perhaps an inset to Figure 1 showing where the three species studied are on a simplified parrot phylogeny would be useful. This is why I hesitated when I saw reference to the cockatiel as it is a member of the cockatoo clade. But I wondered whether reference to the cockatiel (and red and green macaw) had inadvertently been copied in from another manuscript that the authors have prepared.

I have read the paper mainly from the point of view of my interests in the evolution and phylogeny of parrots and not from that of claiming to be a practitioner experienced in modern approaches to karyotype evolution. I therefore found the paper of tremendous interest but had some concerns about whether the authors have over-reached in some of their conclusions.

I attach two files with highlighted suggestions for grammatical and other minor changes.

Reviewer #2:

Remarks to the Author:

The manuscript NCOMMS-21-10399-T studies the genomic rearrangements in parrots using different molecular methods. They found several chromosome rearrangements and gene losses in parrots. Especially transposable elements seem to play an important part in here. The dataset is impressive and the study is interesting also to a wider audience. However, the methods section was really difficult to follow and when comparing the results and methods it was difficult to match the used methods to obtained results. Furthermore, not enough information was given in order to evaluate the analyses and results. For this manuscript to be suitable for publication, extensive re-writing is needed.

General notes

Please write also the scientific names for all the species mentioned in the text.

Please give the whole gene names and symbols when mentioning the genes the first time.

Introduction

Line 64: paris-> pairs

Line 82: please add their divergence time.

Results

Please provide more information about the sequence data generated in this study. How many reads, how many passed quality control etc. Many of that could go to supplementary information.

Line 132: Which two songbirds?

Line 151, parrot-specific gene loss: Did you also check unmapped reads? I couldn't find in the methods how this was done. See for example Laine et al. 2019 BMC Genomics for ideas analyzing the unmapped reads.

Line 181: Which two songbirds?

Lines 206-208: Bit confusing sentence, maybe something like this instead: For example, chr10 was

fused with chr12 in budgerigar, in blue-fronted amazon with chr4a and in monk parakeet with chr6a.

In inter-chromosomal rearrangements, you could also discuss about the relevance of genome assembly quality, how well the chromosomes have been assembled, because that could create some errors in the comparisons.

Lines 243-244: Which five species?

Line 248: PAR explained here and in line 264.

Line 272: Beneficial how?

Conclusions

Line 357: The deleterious effect needs more explanation. Deleterious how? Wouldn't there be selection against it?

Methods

Because you have a long and complicated methods section, a workflow figure would be really helpful in understanding what is happening and in which order. Try also keep the same order both in the methods and the results parts with informative headers. See for example Weissensteiner et al. 2020 in Nature Communications, it has similar methods and that section is well organized with a workflow figure.

Did you do any quality control for the generated and downloaded sequences? Please add that information.

Many programs were missing citations, please cite the programs used.

Cite also data used if downloaded from the repositories like genome version used, NCBI codes, publication citations. These could also be added to the supplementary table.

Lines 370-372: Please provide more information about the Pacbio sequencing. Which method, how many SMRT cells etc.

Re-sequencing: As this isn't part of the genome assembly, I would move this down. And is this the data used in lines 243-244? Where are the three other species?

Lines 385-386: The short-read data, where does this come from? How it was generated?

Lines 390-397: This is confusing. This uses the data generated from the next part? If so, then I would have the order: Pacbio sequencing, Hi-C data generation (the first paragraph of the current "Hi-C data analyses"), Assembly of monk parakeet genome, Assembly of amazon and budgerigar.

Line 399: Wasn't budgerigar also used in Hi-C?

Line 407-415: This is also confusing. Where does it relate to? Please add a better explaining header and an introduction sentence.

Line 426: Where does the RNA-seq data come from? Generated here or already available? Please add more information.

Line 432: Needs a better header. Maybe "Chromosome W linked genes assembly"?

Lines 433-435: Please revise this sentence. How is the great tit related to this section? What data is

that? Which female RNA data did you download?

Lines 465-466: Which genomes were used here?

Line 483, Gene expression analysis: is this related to line 426?

Line 499: Which genomes?

Figures in general, please avoid especially green-red combinations in figures (especially supp.fig 3a). This would help people with color vision deficiency to see the figures properly. Furthermore, if species names aren't mentioned in the figure itself, please mention them in the figure legend. Also explain all the abbreviations used in the figures.

Supplementary Fig. 33 -> Supplementary Fig. 3

Supplementary Fig. 4: Please order the species in same order in figure a and in the species+name+color figure for easier comparison.

Supplementary tables, not sure if it is possible to have those as XLSX-files. In that format the columns wouldn't be cut to several pages like in the pdf-file.

Supplementary table S3: what do the colors mean?

Reviewer #3:

Remarks to the Author:

Comments on Recurrent chromosome reshuffling and the evolution of neo-sex chromosomes in parrots

The present paper analyses an unusual, for being birds, case of multiple chromosomal rearrangements occurring in parrots. The authors have chosen to include of large volume of results which makes evaluating the work difficult (especially as some figures are cartoons and as many figure legends extremely brief). No doubt a lot of data and results are presented, which provides a rich resource for research. However, genomic studies based on only a few species are often descriptive, with support for potential evolutionary processes often being speculative – this is the case also in the present paper. Specifically, the authors promote the suggestion that the loss of 2 specific genes (out of 72 found to have been lost in parrots) has been acting as a driver for genome instability potentially through, or in combination with, sequence repeat accumulation – this is a post-hoc suggestion without statistical support as every genomic feature specific to this clade will appear correlated to the clade's multiple chromosomal rearrangements.

Major comments:

The karyotype and chromosome rearrangements of parrots, including the Monk parakeet, have been studied previously with cytogenetic methods and Furo et al. 2020 is particularly important in this context (this paper is referred to in the present work). Furo et al. provide information about fusions in the Monk parakeet (and other parrot species), and it seems clear that there are both similarities and dissimilarities between these two studies. For example, the fusions between parts of Gga6 and Gga7 forming Monk parakeet chromosome 6b ("Mmo6b") and between parts of Gga6, Gga7 and Gga10 forming Mmo6a (see figure 2) are suggested by both studies. There are also some other similarities between the two studies. However, there are also several discrepancies. For example, Furo et al. suggest that Gga4, Gga14, Gga23 and Gga25 are fused in Monk parakeet, whereas the present work links Gga4 with Gga 13 and Gga15, Gga14 with Gga6 and Gga7, Gga23 with GgaXXX (impossible to distinguish where Gga23 is placed in Monk parakeet in Figure 2), and Gga25 with GgaZ. In particular,

the support for a fusion between Gga4 and Gga25 seems highly convincing in Furo et al., which makes it necessary to gather independent data to verify the link between GgaZ and Gga25 which is presented as a main result here (see e.g. Figure 4f) - e.g. PCR test of multiple male and female Monk parakeets, or new long-read data from independent individuals, would be needed. Furo et al.'s work does not include GgaZ, but the fusion between GgaZ and Gga11 presented here seems robust as it occurs in multiple species and is supported by different coverage values for male and female reads. Finding and describing a new neo-sex chromosome system is valuable as such chromosome translocations have only been described in three bird lineages previously (all referred to here). For the same reason, finding and describing a specific neo-sex chromosome system in birds is not novel either.

The present study is often confirmative rather than novel. The variation in chromosome numbers, and the occurrence of multiple rearrangements with variation among species, is well described previously among parrots (e.g., Furo et al. 2020). Avian neo-sex chromosomes are, as just mentioned, described in several other systems, and multiple chromosome rearrangements is also found before (Kretschmer et al. 2020). The particular value of the present work may thus come from the information in the DNA and RNA sequence data as opposed to the large-scale patterns given by cytogenetics. However, also much of the results from the sequence data is not particularly novel: lineage specific repeat accumulation (Fig. 1a), occurrence of evolutionary strata (Fig. 4c-e), male-biased gene expression (Fig. 5a-c), and variation in repeat accumulation over time on sex chromosomes (Fig. 5d), are well-known from many previous studies in other species. The finding of loss of certain genes in parrots is interesting, and the suggestion that some of them might (ALC1 and PARP3) be related to chromosome instability is an interesting one – and it takes up two sentences in the abstract, suggesting it is take-home message the authors' wish to put forward. However, it is as mentioned above highly speculative at this point. The value of the suggestion would have been much stronger if it could have been backed up with additional data, and the parallel case of multiple chromosome rearrangement occurring in some cuckoo species (Kretschmer et al. 2020) provides such a possibility. Do these cuckoos share the connection between loss of ALC1 and PARP3 and chromosome instability as is now suggested, or is the association between loss of ALC1 and PARP3 and chromosome instability in parrots coincidental or lineage-specific? Similarly, the abstract ends with this final sentence: "Together, the combination of our genomic and cytogenetic analyses highlight the role of TEs and genetic drift in promoting chromosome rearrangements, gene loss and the evolution of neo-sex chromosome in parrots." Nowhere in this study is there evidence of causality, and it is equally likely that TE expansion happened independently of any chromosome change, and selection vs genetic drift is very difficult to infer for the present data. Such statements should be given together with clear disclaimers such as "we speculate that", "this may suggest that" or "it is possible but not proven in this study that".

Moreover, as described on lines 185-186, at least four chromosome rearrangements happened before CRI-psi (an parrot specific TE) expanded, and at the same time as loss of the 72 genes. Thus, contrary to the main conclusions, this suggests if anything that (some of) the rearrangements are not driven by these events and it is at least not in line with TE driving rearrangements. Furthermore, as it is at the same time speculated that the loss of ALC1 was caused/driven by the CRI-psi expansion (Fig. 1b) this also weakens the argument that the ALC1-loss would be associated with (drive) chromosome rearrangements.

The authors repeatedly stress that these chromosome rearrangements are rapidly occurring, but in fact some of these fusions occurred at least 37 Myr ago, and the three main study species split c. 20 Myr ago, so there have been plenty of evolutionary time for any of these rearrangements to occur. For example, these dates precede most (all?) speciation events among Hominoidea (including gibbons, orangutans, gorilla, chimp and us). I would therefore not describe this as rapid evolution (it is rather so that avian chromosome evolution is extremely slow). If the authors wish to study rapid evolution, they need to include additional species or sub-species with much more recent speciation history than these three species splitting 20 Myr ago.

Specific comments:

Line 88: Remove "R."

Line 91: An example of uncaredful interpretation – neither of these three studies reveal this, but they are all in line with this suggestion. There are also several studies that are not in line with adaptive evolution of Y and they should also be highlighted.

Line 97: "recently". This study is as recent as many of the studies in the sentence above.

Line 100: What does "re-produce" mean?

Line 101: "reveals". This is already known and well-described – use "confirm" instead.

Line 111: PacBio or Nanopore? Give info in parenthesis.

Line 114: Good!

Line 118-119: Even the latest zebra finch W (i.e. a species without any translocation) is longer than that (c. 20 Mb). Also, 13.8 Mb is surprisingly short as the W seems almost as long as the Z (which should be over 80 Mb) on the karyotypes (shown in this and other papers). What does this say about the quality of the W chromosome assembly presented in this paper? More details of the W chromosome are needed.

Line 138. This is then not recent events - see comment above.

Line 146-149: This result is presented in Figure 1a and is very difficult to understand as there is no explanation in the figure. Please expand this section here and provide understandable information in the figure (e.g. a y-axis with a label telling exactly what the vertical bars are showing). Also, are there also novel TEs in the other lineages presented? By only plotting parrot-specific TEs this illustration will look like this is something unique for parrots which may not be the case?

Line 160: This is very important information for the idea the authors promote throughout the study. Thus, rather than just listing a lot of papers (many studies does not necessarily prove anything; do we even know whether they are independent?) explicit results from these studies should be given (what did they show, in what species, is it relevant to birds, etc.) – ideally in the main text as well as supplementary text.

Line 165: What does "seems to be associated with" mean? Figure 1b clearly wishes to show this.

Line 173-174. This should/can be tested statistically using the position of the 72 genes.

Line 189: Give explicit counts rather than inexact information such as "most".

Line 191: Give time estimate in Myr rather than use information such as "recent". Additional species (speciation event) need to be included in order to achieve better resolution on timing.

Line 212-213: This section needs better motivation. What is this hypothesis suggesting in terms of adaptive evolution and is there any previous study supporting it? It is not well-described how the insulation scores relate to Guerrero and Kirkpatrick's hypothesis. What is an evolutionary breakpoint and how does it relate to fissions? Perhaps the section can be given as supplement or at least the figure?

Line 233. Again give time estimates rather than vague terms such as newly.

Line 228-229: This is a very interesting result, novel to this study! But see comment above regarding Chr25.

Line 233: Check chromosome numbers and chromosome lengths. Gan et al. show that Chr1A is fused to Z.

Line 239, 245: This also suggests a relatively old age of this chromosome fusion.

Line 241: See comment above about Chr25.

Line 251-253. This is now revealed in this study, right? I believe you followed the previous definitions.

Line 257. The use of homologous ZW gene pairs here, suggest that the ZW chromosomes is not hemizygous as written above (line 239).

Line 260. Where do you show that this is a single stratum? I believe you need to evaluate the timing of recombination suppression along the fused region, before defining it as a single stratum.

Line 263. Do you show that Chr11 is attached to both Z and W?

Line 267-268. The data in Fig. 4e is from a single gametolog at Chr11. This confirms that the fusion has this age and that recombination stopped at that particular position, but as mentioned just above this does not necessarily mean that this was the case for the whole region.

Line 283. I do not follow "Additionally, W-linked gene loss leads to imbalanced dosage of the proto-sex-linked genes." Additionally to what? Imbalance of what?

Line 286. "...fully..." Does this mean that your result suggest that S4 has evolved a mechanism to partly compensate? I think you should give data on mean \pm SD in different groups rather than using loose descriptions such as not fully.

Line 288, 292. Again, you have very little data on the rate of gene loss and dosage compensation (see comments above); thus, your study does not provide such a unique window into the temporal dynamics as now stated.

Line 297-316. How does this large size relate to you estimate of the W being 13.2 Mb (lines 117-118)?
Line 302 – yes, repeat accumulation is also shown in several other avian W chromosomes. The discrepancy regarding the location of Chr25 between Furo et al. 2020 and this study needs to be clarified, before any further speculation. Also, even though the proportion of satellite repeats is higher on Chr25 in 2 of 3 species presented, there is also satellite repeats on all other chromosomes that could have acted as a source of satellite repeat expansion on the W.

Line 323-324. I'm not sure why speciation is discussed here?

Line 325. "...slightly..." What does slightly mean here? Is this data shown and tested?

Line 326. "...speculate...". I like such careful wording regarding the conclusions in the discussion.

Line 343. TAD is not defined before. Also, it is not explained why this would be genetic drift rather than natural selection. Fission placements is a mechanism and not a process.

Line 344. "One of the consequences...". There are cases of neo-sex chromosome formation in e.g. mammals and birds without additional inter-chromosomal rearrangements, so this is not a general requirement. Moreover, also in the present study it is premature to draw this conclusion as the fusion

involving Chr11 is ancient within parrots and could have been the first to appear (i.e. not necessarily a consequence of frequent inter-chromosomal rearrangements (arguably, it is even impossible to exclude the alternative, i.e. that the fusion involving Chr11 was driving additional inter-chromosomal rearrangements)).

Line 363-523. In general, more information regarding the methods (and detailed results) would be desirable.

Figure 1a is beautiful but difficult to follow from the figure per se and the legends. A more informative figure description would be preferable.

Figure 2 is very nice, but some connections are difficult to follow. Details should be given as supplement.

Figure 3. The content of this figure is particularly difficult to condense (as is the result and its implication for chromosome evolution).

Figure 4. Very nice and informative panel. However, gene trees (4e) should be given also for other gametologs along the chromosome. And as mentioned above the chr25-discrepancy to Furo et al. needs to be evaluated. 4f – do you have data explicitly showing that both 11 and 25 are physically linked to both Z and W (this does not need to be the case as e.g. Z1Z2W is also possible).

Figure 5a is very difficult to interpret (more information in the legend and possible in the figure would be desirable).

Figure 6. This figure shows a minor result (better as a supplement?).

Reviewer #1 (Remarks to the Author):

This paper offers a potentially fascinating findings about the frequency and extent of chromosomal changes in parrots. I found some of the statements about how how specific some of the chromosomal changes are to one or other lineage gave me pause. I wondered whether more extensive clade sampling of the parrot phylogeny would dictate modifications to those statements. Perhaps an inset to Figure 1 showing where the three species studied are on a simplified parrot phylogeny would be useful. This is why I hesitated when I saw reference to the cockatiel as it is a member of the cockatoo clade. But I wondered whether reference to the cockatiel (and red and green macaw) had inadvertently been copied in from another manuscript that the authors have prepared.

R: Thanks for the reviewer's comments. In this revised manuscript, we try to not stress that some events are specific to certain lineages, but instead to suggest that the karyotype changes occurred in the ancestral lineage leading to the species (but do not know the exact time point). We agree that sampling more species will certainly help pinpoint more precisely the time points the karyotype changes took place, but it should not change the observations that most karyotype changes are not shared by species. This is the case when we added a Kakapo (a deep branching parrot) genome that was recently published by the Vertebrate Genome Project (VGP) (Rhie et al. 2021 Nature).

Among the parrot species in Figure1, only three (now four after we added kakapo) have chromosome-level assemblies. In the revised figure, we have highlighted these four parrots by adding red asterisks next to them and indicated in the legend that they are the species studied for their karyotype evolution.

Regarding cockatoo (this also replies to the comment in the Reporting Summary), we did sequence two additional parrot species (red-and-green macaw and cockatiel) with low coverage. This is mentioned in Line 243-246, and the results are shown in the supplementary figure S9 (so they are not for another manuscript). The low-coverage sequencing data of these species does not permit a genome assembly, but it is useful to confirm that neo-sex chromosomes involving chr11 also occurred in other parrots including cockatoo. To clarify, we have shown the species names of the five parrots at line 243-246:

"Re-sequencing data from five additional parrot species: Carolina parakeet (*Conuropsis carolinensis*), Hispaniolan amazon (*Amazona ventralis*), Puerto Rican amazon (*Amazona vittata*), Red-and-green macaw (*Ara chloropterus*) and Cockatiel (*Nymphicus hollandicus*) (Supplementary Fig. S11), and a new chromosome-level assembly of the deep-branching Kakapo (Rhie et al. 2021) further support that the fusion of chr11 into sex chromosomes is shared by all parrots", as well as in the supplementary figure S11. For the other three parrots, we used the published resequencing data, details are explained in the legend of the supplementary figure S11.

I have read the paper mainly from the point of view of my interests in the evolution and phylogeny of parrots and not from that of claiming to be a practitioner experienced in modern approaches to karyotype evolution. I therefore found the paper of tremendous interest but had some concerns about whether the authors have over-reached in some of their conclusions.

R: Thank you for the positive remarks on this manuscript. We agree that we may have overemphasized the causal roles of transposable elements or gene loss, and have accordingly toned down the claim by saying in the abstract “suggests TEs and genetic drift may involve in chromosomal rearrangements and the evolution of neo-sex chromosome in parrots.” We have also toned down any claims about the fast rate of the chromosomal changes (as other reviewers have pointed out) since we only sampled four genomes of parrots.

I attach two files with highlighted suggestions for grammatical and other minor changes.

R: We have replied to the comments in the files the reviewer attached.

Reviewer #2 (Remarks to the Author):

The manuscript NCOMMS-21-10399-T studies the genomic rearrangements in parrots using different molecular methods. They found several chromosome rearrangements and gene losses in parrots. Especially transposable elements seem to play an important part in here. The dataset is impressive and the study is interesting also to a wider audience. However, the methods section was really difficult to follow and when comparing the results and methods it was difficult to match the used methods to obtained results. Furthermore, not enough information was given in order to evaluate the analyses and results. For this manuscript to be suitable for publication, extensive re-writing is needed.

R: Thanks for the positive remarks. We have extensively re-written the methods part, and added more details in the results and figure legends.

General notes

Please write also the scientific names for all the species mentioned in the text.

R: We have added the scientific names for all species when are mentioned for the first time.

Please give the whole gene names and symbols when mentioning the genes the first time.

R: We have added the whole gene names for *ALC1* and *PARP3*.

Introduction

Line 64: paris-> pairs

R: Revised

Line 82: please add their divergence time.

R: We have added the divergence time of less than 30 million years according to one study in 2012 (10.1111/j.1463-6409.2012.00567.x).

Results

Please provide more information about the sequence data generated in this study. How many reads, how many passed quality control etc. Many of that could go to supplementary information.

R: We have added the information of sequencing coverage at line 736, line 658, line 647. Unfortunately we do not have information about the proportion of passed reads since what we received from the sequencing company (Annoroad) were already “cleaned reads” that can be directly used.

Line 132: Which two songbirds?

R: We have rewritten this sentence by listing the names of the two songbirds.

Line 151, parrot-specific gene loss: Did you also check unmapped reads? I couldn't find in the methods how this was done. See for example Laine et al. 2019 BMC Genomics for ideas analyzing the unmapped reads.

R: We did not check the “unmapped reads”, mainly because, unlike short-read based assemblies, long-read genome assemblies contain much fewer gaps and most reads should have been used to produce contig sequences. In the case of *ALC1*, for instance, at the locus where *ALC1* was (partially) deleted, the assembled sequence contains the nearby genes in a single contig (gap-free), and the partial sequences of *ALC1* indicated that multiple deletion events have led to *ALC1* pseudogenization. Unfortunately, it's impossible to manually examine every case in the way that has been done for *ALC1*. Given that we have examined the transcriptomes from nine different tissues of both sexes (so 18 transcriptomes, each with three biological replicates), it should be very unlikely we failed to assemble those genes.

Line 181: Which two songbirds?

R: Added the names of the two songbirds.

Lines 206-208: Bit confusing sentence, maybe something like this instead: For example, chr10 was fused with chr12 in budgerigar, in blue-fronted amazon with chr4a and in monk parakeet with chr6a.

R: Revised according to the reviewer's suggestion.

In inter-chromosomal rearrangements, you could also discuss about the relevance of genome assembly quality, how well the chromosomes have been assembled, because that could create some errors in the comparisons.

R: Thanks for the good suggestion. We added in the Discussions: “Though some microchromosomes were missing in the genome assemblies of blue-fronted amazon and budgerigar, they represent a tiny proportion of the genomes, and can only lead to an underestimation of the frequency of chromosomal changes”. Please note we only compare those genomes with chromosome-level assemblies, and to reduce the bias due to different scaffolding pipelines, we re-analyzed the Hi-C data of budgerigar and used the same method we used for monk parakeet and blue-fronted amazon. For all chromosomal models, we have

done manual curations to make sure there were no major errors in the assembly of chromosomal models.

Lines 243-244: Which five species?

R: The species names are all listed now.

Line 248: PAR explained here and in line 264.

R: We have removed the redundant explanation.

Line 272: Beneficial how?

R: We added “acquired female-specific or female-biased expression”.

Conclusions

Line 357: The deleterious effect needs more explanation. Deleterious how? Wouldn't there be selection against it?

R: We added that the accumulation of satellite DNA and heterochromatin can cause “genome instability”, and explained that “The lack of recombination of the W chromosome made it ineffective in purging the deleterious mutations”.

Methods

Because you have a long and complicated methods section, a workflow figure would be really helpful in understanding what is happening and in which order. Try also keep the same order both in the methods and the results parts with informative headers. See for example Weissensteiner et al. 2020 in Nature Communications, it has similar methods and that section is well organized with a workflow figure.

R: Thanks for the helpful suggestions. We have added one supplementary figure (a workflow, Fig. S1) explaining the sources of the chromosome-level genomes. We have also re-organized this section according to the reviewer's suggestions.

Did you do any quality control for the generated and downloaded sequences? Please add that information.

R: For long-reads, we took all the reads but discarded those shorter than 8000 bp. This was done by the assembler. We have added this information. Note we have uploaded the full configuration file to Github, so we added: “The full configuration has been deposited to Github (see Code Availability)”. For Hi-C reads, we did not trim the reads because the soft-clipping alignment was used anyway. Similarly, for RNA-seq data we were only concerned about the expression levels, so trimming was not required and soft-clipping alignment was applied. For re-sequencing data, we did not trim the reads either because it would not have a great effect on calculating sequencing coverage or calling variants. Throughout the manuscript, we added the indication that we used the “raw reads”.

Many programs were missing citations, please cite the programs used.

R: We have added citations for the programs throughout the method part.

Cite also data used if downloaded from the repositories like genome version used, NCBI codes, publication citations. These could also be added to the supplementary table.

R: We have added the Supplementary table S6 to include this information.

Lines 370-372: Please provide more information about the Pacbio sequencing. Which method, how many SMRT cells etc.

R: We have added more details about PacBio sequencing at line 645-648:

“The extracted DNA was used to construct a 20-kb PacBio SMRTbell™ library prepared with the Sequel Sequencing Kit 3.0, according to the released protocol from the PacBio Company. The library was sequenced on the Sequel II machine, producing ~80 Gb reads in one SMRT cell.”

Re-sequencing: As this isn't part of the genome assembly, I would move this down. And is this the data used in lines 243-244? Where are the three other species?

R: Yes those low-coverage data are not for genome assembly, but for confirming the neo-sex chromosome in other parrot species. We have moved this part after the sex chromosome section. We added an explanation of how we used these species at line 451-456:

“Re-sequencing data from five additional parrot species: Carolina parakeet (*Conuropsis carolinensis*), Hispaniolan amazon (*Amazona ventralis*), Puerto Rican amazon (*Amazona vittata*), red-and-green macaw (*Ara chloropterus*) and cockatiel (*Nymphicus hollandicus*) (Supplementary Fig. S11), and a new chromosome-level assembly of the deep-branching kakapo (Rhie et al. 2021) further support that the fusion of chr11 into sex chromosomes is shared by all parrots”

Lines 385-386: The short-read data, where does this come from? How it was generated?

R: The short-read data come from the same individual we sequenced for long-reads. We have moved this into a new section “short-read sequencing” and added more details.

Lines 390-397: This is confusing. This uses the data generated from the next part? If so, then I would have the order: Pacbio sequencing, Hi-C data generation (the first paragraph of the current “Hi-C data analyses”), Assembly of monk parakeet genome, Assembly of amazon and budgerigar.

R: Yes, the Hi-C data was from the next section. We have now re-organised the method sections to have “long-read sequencing”, “short-read sequencing”, “Assembly of monk parakeet genome”, “Hi-C scaffolding”, “Chromosomal assembly of blue-fronted amazon and budgerigar”, reflecting the order of our workflow.

Line 399: Wasn't budgerigar also used in Hi-C?

R: That's right, we have included this information in a new section “chromosomal assembly of blue-fronted amazon and budgerigar”. We have also added a workflow in the supplementary figure S1 to illustrate how each chromosome-level genome was generated.

Line 407-415: This is also confusing. Where does it relate to? Please add a better explaining header and an introduction sentence.

R: That part is related to the analysis of genome foldings. We have now added a new introduction sentence “We calculated insulations scores in order to investigate whether evolutionary breakpoints tend to locate at TAD boundaries.”

Line 426: Where does the RNA-seq data come from? Generated here or already available? Please add more information.

R: The RNA-seq data is generated in this study. Those RNA-seq data were generated to compare the expression levels of sex-linked genes between the males and females, but they were also useful for gene annotations. We have added a section called “sample collections” as the first part of Methods where we also directed the reader to the Supplementary table S2 that contains all information about samples we collected.

Line 432: Needs a better header. Maybe “Chromosome W linked genes assembly”?

R: We have integrated this part into the “evolutionary strata” part because the gametologs were assembled in order to confirm the evolutionary strata we have assigned.

Lines 433-435: Please revise this sentence. How is the great tit related to this section? What data is that? Which female RNA data did you download?

R: We have moved this part into the “evolutionary strata” and re-written it, as well as adding a citation:

“For great tit and budgerigar, since the genomes were derived male individuals, we downloaded the female RNA-seq data (SRR2170826 and SRR5336544 (Cooke et al. 2017) respectively) from NCBI SRA to assemble the transcriptomes using Trinity (2.8.4) (Haas et al. 2013), in order to assemble the sequences of W-linked genes.”

Lines 465-466: Which genomes were used here?

R: Here we used the chromosome-level genomes as seen in Figure 2. We added “see Figure 2 for the alignment pairs”.

Line 483, Gene expression analysis: is this related to line 426?

R: Yes, the data we mentioned in line 426 come from here. We have changed the title to “sex chromosome gene expression” as we are only concerned about the expression of sex-linked genes. Those RNA-seq data were also used for genome annotations. In the first Method section “sample collections” we have added an explanation as to what the RNA-seq data was generated for.

Line 499: Which genomes?

R: Added “monk parakeet”.

Figures in general, please avoid especially green-red combinations in figures (especially supp.fig 3a). This would help people with color vision deficiency to see the figures properly.

Furthermore, if species names aren't mentioned in the figure itself, please mention them in the figure legend. Also explain all the abbreviations used in the figures.

R: We have changed the colors in supp.fig 3a and supp.fig 3b (now 4a and 4b). We have also added species names and explanations for abbreviations in the legend.

Supplementary Fig. 33 -> Supplementary Fig. 3

R: Typo corrected.

Supplementary Fig. 4: Please order the species in same order in figure a and in the species+name+color figure for easier comparison.

R: We have placed the species in the same order. We have also added kakapo whose genome was recently published (Rhie et al. 2021 Nature).

Supplementary tables, not sure if it is possible to have those as XLSX-files. In that format the columns wouldn't be cut to several pages like in the pdf-file.

R: We would also prefer XLSX files. We will discuss with the editor for options.

Supplementary table S3: what do the colors mean?

R: At the end of the table we have added a legend indicating the two colors highlight the 5' and 3' flanking genes respectively.

Reviewer #3 (Remarks to the Author):

Comments on Recurrent chromosome reshuffling and the evolution of neo-sex chromosomes in parrots

The present paper analyses an unusual, for being birds, case of multiple chromosomal rearrangements occurring in parrots. The authors have chosen to include a large volume of results which makes evaluating the work difficult (especially as some figures are cartoons and as many figure legends extremely brief). No doubt a lot of data and results are presented, which provides a rich resource for research. However, genomic studies based on only a few species are often descriptive, with support for potential evolutionary processes often being speculative – this is the case also in the present paper. Specifically, the authors promote the suggestion that the loss of 2 specific genes (out of 72 found to have been lost in parrots) has been acting as a driver for genome instability potentially through, or in combination with, sequence repeat accumulation – this is a post-hoc suggestion without statistical support as every genomic feature specific to this clade will appear correlated to the clade's multiple chromosomal rearrangements.

R: Many thanks for the critical and helpful comments that we do believe can help improve the manuscript. We fully agree with the reviewer that some proper legends are lacking, and during the revision, we have added more details in the legends. Regarding the “cartoons”, I believe the reviewer referred to Fig. 4a in which we used a schematic diagram to illustrate the evolutionary history that should be very helpful for readers to understand the order of evolutionary events.

We also agree that genomic studies sometimes are descriptive, and some of the conclusions are speculative; in the case of the present study, we hypothesized that two genes (*ALC1* and *PARP3*) may be potentially responsible for frequent chromosomal rearrangements in parrots. Importantly, however, this hypothesis is testable (though we have not tested it in this paper), for instance, one could knock out these two genes using the CRISPR-Cas9 technique in chicken cell lines to examine the functional impact of the loss of *ALC1* and *PAPR3*. This test wouldn't be possible without us having identified the two lost genes (after literature reviews for all 72 lost genes) in the present study. The reviewer is right that some genomic features are simply correlated with chromosomal rearrangements, but we believe the present study is also valuable by putting forward new hypotheses to be tested in future studies. That being said, we have carefully gone through the comments by all three reviewers, and rewritten some of the sentences to tone down the claims.

Major comments:

The karyotype and chromosome rearrangements of parrots, including the Monk parakeet, have been studied previously with cytogenetic methods and Furo et al. 2020 is particularly important in this context (this paper is referred to in the present work). Furo et al. provide information about fusions in the Monk parakeet (and other parrot species), and it seems clear that there are both similarities and dissimilarities between these two studies. For example, the fusions between parts of Gga6 and Gga7 forming Monk parakeet chromosome 6b (“Mmo6b”) and between parts of Gga6, Gga7 and Gga10 forming Mmo6a (see figure 2) are suggested by both studies. There are also some other similarities between the two studies. However, there are also several discrepancies. For example, Furo et al. suggest that Gga4, Gga14, Gga23 and

Gga25 are fused in Monk parakeet, whereas the present work links Gga4 with Gga 13 and Gga15, Gga14 with Gga6 and Gga7, Gga23 with GgaXXX (impossible to distinguish where Gga23 is placed in Monk parakeet in Figure 2), and Gga25 with GgaZ. In particular, the support for a fusion between Gga4 and Gga25 seems highly convincing in Furo et al., which makes it necessary to gather independent data to verify the link between GgaZ and Gga25 which is presented as a main result here (see e.g. Figure 4f) - e.g. PCR test of multiple male and female Monk parakeets, or new long-read data from independent individuals, would be needed. Furo et al.'s work does not include GgaZ, but the fusion between GgaZ and Gga11 presented here seems robust as it occurs in multiple species and is supported by different coverage values for male and female reads. Finding and describing a new neo-sex chromosome system is valuable as such chromosome translocations have only been described in three bird lineages previously (all referred to here). For the same reason, finding and describing a specific neo-sex chromosome system in birds is not novel either.

R: Many thanks for the detailed comments. We are fully aware that karyotype evolution has been studied with cytogenetic methods in parrots (as in most other bird clades (Kretschmer et al. 2018 Genes)). Importantly, Furo et al. 2020 and Furo et al. 2017 actually inspired this present study to employ genomic methods (Note both Dr. Furo and Dr. de Oliveira are co-authors of this manuscript and have contributed to verifying the chromosome changes we identified through genomic assembly), for the following reasons:

1. Some chromosomes are difficult to be differentiated morphologically, but once the sequences are assembled into chromosome models, the chromosomes can be identified through sequence alignment with the chicken chromosomes. For instance, Furo et al. (2020) were aware that Gga25 fused with a large chromosome but that chromosome was wrongly identified as Gga4 which was actually GgaZ. We (together with Dr. Furo and Dr. de Oliveria) have re-visited the previous results, and now the patterns make sense after combining the genome assembly data.

2. Most cytogenetic approaches have limited resolution in studying microchromosomes (particularly smaller ones) in birds. And again the small microchromosomes are morphologically difficult to be identified. In the present study, we have learned that microchromosomes can frequently fuse to macrochromosomes in, for example, monk parakeet, and many of the fusions have not been identified in Furo et al 2020. Moreover, we identified several cases where new microchromosomes were created through fissions. Those newly formed microchromosomes will be useful models to study how the distinct genomic features of microchromosomes evolve, and the fused microchromosomes can be good models to study, for example, whether and how their genomic features change following their fusion with macrochromosomes.

3. We were now able to study the temporal evolution of sex chromosome differentiation with molecular data thanks to the assembled sequences of the Z and W chromosomes. Because Gga11 fused with the Z and W chromosome in the ancestor of parrots more than 37 million years ago, the neo-W is substantially degenerated, becoming distinct from the neo-Z, so it has been a challenge to use the probe of chicken Gga11 to hybridize with the neo-W. In the present study, we have annotated genes (gametologs) in both neo-W and neo-Z, confirming that Gga11 fused with both Z and W, and also provide detailed information including how many genes were lost and which genes were maintained in the neo-W since the Gga11-ZW fusion, which would have been impossible to be obtained through cytogenetic study. Moreover, the

chromosome-level assembly has the power to identify the fusion of Gga25 into the sex chromosomes which seems obscured in the results of Furo et al. 2020.

4. Furo et al. (2017) attempted to identify repeat sequences that are responsible for the enlargement of the W chromosome in monk parakeet, but was limited to a few satellite probes ((GGA)_n for instance). Through genome sequencing, we identified an abundant 20-bp satellite DNA on the W chromosome which allows for designing new probes for validation. This 20-bp satellite was almost impossible to be identified through cytogenetic methods alone.

For the reasons given above, we believe the Furo et al. 2020 study was at best incomplete, not to mention that some chromosomes were incorrectly recognized. We believe the present work sets an exciting example where genomic and cytogenetic methods complement each other and address scientific questions that are difficult to be addressed with genomic or cytogenetic tools alone, and we predict combining genomics and cytogenetics will be a new norm in future study of karyotype evolution.

As for the specific comments on the disparity between this work and Furo et al. 2020, as we mentioned above, GgaZ was wrongly identified as Gga4. Dr. Furo has re-done this experiment (Supplementary Fig. S10a), realizing that the large chromosome which Gga25 fused is GgaZ, but not Gga4.

Regarding the finding of neo-sex chromosome in birds, it is true that neo-sex chromosomes have been reported in other bird lineages, but has not been reported in parrots before. Our finding that Gga11 and Gga25 sequentially fused with the sex chromosomes in monk parakeet enriches the study of avian neo-sex chromosome; moreover, our assembly using long-reads allows characterization of the evolutionary pattern and processes of neo-sex chromosomes at the molecular level (gene loss, genetic decay, gene expression, evolutionary strata, dosage compensation, etc.).

The present study is often confirmative rather than novel. The variation in chromosome numbers, and the occurrence of multiple rearrangements with variation among species, is well described previously among parrots (e.g., Furo et al. 2020). Avian neo-sex chromosomes are, as just mentioned, described in several other systems, and multiple chromosome rearrangements is also found before (Kretschmer et al. 2020). The particular value of the present work may thus come from the information in the DNA and RNA sequence data as opposed to the large-scale patterns given by cytogenetics. However, also much of the results from the sequence data is not particularly novel: lineage specific repeat accumulation (Fig. 1a), occurrence of evolutionary strata (Fig. 4c-e), male-biased gene expression (Fig. 5a-c), and variation in repeat accumulation over time on sex chromosomes (Fig. 5d), are well-known from many previous studies in other species. The finding of loss of certain genes in parrots is interesting, and the suggestion that some of them might (ALC1 and PARP3) be related to chromosome instability is an interesting one – and it takes up two sentences in the abstract, suggesting it is take-home message the authors' wish to put forward. However, it is as mentioned above highly speculative at this point. The value of the suggestion would have been much stronger if it could have been backed up with additional data, and the parallel case of multiple chromosome rearrangement occurring in some cuckoo species (Kretschmer et al. 2020) provides such a possibility. Do these cuckoos share the connection between loss of ALC1 and PARP3 and chromosome instability as is now suggested, or is the association between loss

of ALC1 and PARP3 and chromosome instability in parrots coincidental or lineage-specific? Similarly, the abstract ends with this final sentence: "Together, the combination of our genomic and cytogenetic analyses highlight the role of TEs and genetic drift in promoting chromosome rearrangements, gene loss and the evolution of neo-sex chromosome in parrots." Nowhere in this study is there evidence of causality, and it is equally likely that TE expansion happened independently of any chromosome change, and selection vs genetic drift is very difficult to infer for the present data. Such statements should be given together with clear disclaimers such as "we speculate that", "this may suggest that" or "it is possible but not proven in this study that".

R: As we explained in the response to the last comment, the findings in Furo et al. 2020 were at best incomplete (very little information about intra-chromosomal rearrangements, incomplete information about microchromosome, and sometimes the inability to differentiate translocations from fusions) and sometimes misidentified chromosomes (in the case of the suggested Gga25+Gga4 fusion for instance). In the present study, with the whole-genome sequences of monk parakeet assembled into 24 chromosome models (the known haploid number is 24), we are able to reveal a much more complete picture of chromosomal rearrangements, and identified two events of autosome-sex chromosome fusions, none of which were identified in Furo et al. 2020.

Regarding neo-sex chromosomes in parrots, we would like to remind the reviewer that our study focuses on the neo-sex chromosomes, not the old sex chromosomes that have been extensively studied as the reviewer pointed out. In particular, we collected transcriptomes from nine female tissues and nine male tissues (each has three replicates), aiming to address whether the forming of neo-sex chromosome was driven by female-specific selection, that is, whether the fusion of the Gga11 with the ZW chromosome is favored by females because Gga11 may contain female-beneficial genes which would favor their linkage with the female-specific W chromosomes. The lack of enrichment of female-biased genes, rapid accumulation of transposable elements, a lack of complete global dosage compensation, on the neo-W chromosome, all suggest that the neo-W is going through a degenerative process, instead of evolving a repertoire of female-biased genes. On top of that, the neo-sex chromosome also provides a new opportunity to study the evolutionary processes at an early stage of sex chromosome differentiation. As illustrated in Fig. 4f, the evolutionary stratum of the neo-sex chromosome is younger than the youngest evolutionary stratum of the old sex chromosome, and we provided some new insights into the temporal patterns of transposable element accumulation and gene loss on the neo-sex chromosome.

As for other comments about novelty:

lineage specific repeat accumulation (Fig. 1a): We agree that this happens in many bird lineages, for instance, increased LTR activity in passerines, and CR1 activity in chicken. In the present study, we actually identified a novel parrot-specific CR1 element (CR1-psi) that account for as much as one-third of total TE content in parrots, and estimated their abundance and timing. This is a novel finding and it would not have been uncovered if we did not compare multiple genomes of parrot and non-parrot birds. Further study to illustrate the impact and regulatory role of CR1-psi in parrots wouldn't be possible without our identification of CR1-psi in this study.

occurrence of evolutionary strata (Fig. 4c-e): We agree that it is known that evolutionary strata occurred several times on the sex chromosomes of birds, including songbirds (Zhou et al. 2014; Xu et al. 2019). What is unknown is whether parrots have a lineage-specific stratum. Using sequence divergence between the Z and W chromosome, we showed that parrot has an evolutionary stratum that evolved after its split from songbirds. This parrot-specific stratum was further supported by the gametolog phylogeny. Furthermore, we characterized the patterns of evolutionary stratum of the neo-sex chromosome in parrots.

male-biased gene expression (Fig. 5a-c), and variation in repeat accumulation over time on sex chromosomes (Fig. 5d): As we responded above, we focused here on the neo-sex chromosomes. The evolution of dosage compensation of repeat accumulation has not been intensively studied for bird neo-sex chromosomes, thus our work provides novel findings for neo-sex chromosomes.

Regarding *ALC1* and *PAPR3*, it should be noted that they were investigated only after we had closely examined the literature on all 72 lost genes. We should, however, as the reviewer suggested below, explicitly provide more information about these two genes, not just listing the literature (indeed a substantial number of studies have been done independently on *PAPR3* and *ALC1*). In birds of prey where frequent chromosomal changes have also been observed, we have detected intact copies of *ALC1* and *PAPR3*. It would have been striking should the birds of prey lost *ALC1* and *PAPR3* in parallel. A more likely scenario, in our view, would be that another genetic mechanism in birds of prey has convergently contributed to genome instability. To test whether the loss of *ALC1* and *PAPR3* may have contributed to chromosomal changes, from our point of view, a more direct approach would be needed to examine the consequence of knockout of *ALC1* and *PAPR3* in the cell lines of chicken or zebra finch; but unfortunately, this takes a large amount of time and is beyond the scope of the present study. During this revision, we carefully reviewed the literature on *ALC1* and *PAPR3* again, and wrote in the main text about what we already know about the functions of these two genes; moreover, we further added in the Discussion that more study is needed to test for a causal role of *ALC1* and *PAPR3* on chromosomal rearrangements: “However, the impact of the loss of *ALC1* and *PAPR3* needs to be experimentally investigated to test for their role in promoting chromosomal rearrangements and genome instability,”

Regarding the last sentence in the abstract, we agree it has been overstated. In this revision, we replaced this sentence with “suggests TEs and genetic drift may involve in chromosomal rearrangements and the evolution of neo-sex chromosome in parrots.”

Moreover, as described on lines 185-186, at least four chromosome rearrangements happened before CRI-psi (an parrot specific TE) expanded, and at the same time as loss of the 72 genes. Thus, contrary to the main conclusions, this suggests if anything that (some of) the rearrangements are not driven by these events and it is at least not in line with TE driving rearrangements. Furthermore, as it is at the same time speculated that the loss of *ALC1* was caused/driven by the CRI-psi expansion (Fig. 1b) this also weakens the argument that the *ALC1*-loss would be associated with (drive) chromosome rearrangements.

R: We thank the reviewer for pointing out this concern. We fully agree that some of the associations reflect correlation but not causation. As in this case, we would not intend to

suggest that all events of chromosomal rearrangements were caused by CR1-psi activities, but very likely some of the events were linked to the loss of *ALC1* and *PARP3*. We also do not intend to suggest that the loss of all 72 genes in parrots was driven by the activity of CR1-psi. We would like to point out, however, we had limited power to determine the precise order of evolutionary events of CR1-psi emergence, and loss of *ALC1* and *PARP3* and chromosomal changes. In particular, CR1-psi originated in the lineage leading to parrots between 37 (the common ancestor of parrots) to 65 (the common ancestor of parrots and passerines) million years ago, and at some point during this period, *ALC1* and *PARP3* were deleted - therefore we were unable to determine which event preceded the other. What is true, however, is that most (32 out of 38) chromosomal rearrangements occurred following the ancestral loss of *ALC1* and *PARP3* (only six chromosome rearrangements might have happened before CR1-psi origination, but again we do not know the relative order). Accordingly, we toned down some of the claims and suggested we only observed correlational relationships between CR1-psi expansion and chromosomal rearrangements.

The authors repeatedly stress that these chromosome rearrangements are rapidly occurring, but in fact some of these fusions occurred at least 37 Myr ago, and the three main study species split c. 20 Myr ago, so there have been plenty of evolutionary time for any of these rearrangements to occur. For example, these dates precede most (all?) speciation events among Hominoidea (including gibbons, orangutans, gorilla, chimp and us). I would therefore not describe this as rapid evolution (it is rather so that avian chromosome evolution is extremely slow). If the authors wish to study rapid evolution, they need to include additional species or sub-species with much more recent speciation history than these three species splitting 20 Myr ago.

R: It's true that some of the chromosomal rearrangements have an ancient origin, for instance, those occurred at the ancestor of all parrots 37 Myr ago, but it was not our intention to suggest that ALL chromosomal rearrangements are recent. In other words, having observed some ancient chromosome changes does not defy the notion that in some parrot lineages chromosomal changes can be recent. As we wrote in line 561-562 "suggesting karyotypic changes have been regularly taking place in the course of parrot diversification". Though we only sampled three parrots (now four after we added kakapo), we do know that many chromosomal rearrangements are not ancient events. For instance, the chromosomal events that are specific to the lineage leading to monk parakeet must have occurred after its split from the blue-fronted amazon lineage. That is, 4 fissions and 9 fusions occurred within the last 20 Myr in the lineage leading to monk parakeet. If we could calculate it in a naive way (of course this is not a scientifically appropriate way because we did not sample any sub-species in this lineage), this is equivalent to 0.65 chromosomal changes per Myr. This is extraordinarily striking given most other birds have an extremely slow rate of karyotype evolution as the reviewer pointed out. In this context, we consider at least for monk parakeet the change of karyotype has been at a relatively fast rate. That being said, during this revision, we toned down any claims about the rate of chromosomal rearrangements, but rather emphasize the frequency and independent occurrence of chromosomal rearrangements to avoid confusion or misunderstanding.

It should be pointed out here (also mentioned above) that in this revision we included one more parrot, kakapo, that has a chromosome-level assembly and was recently published (Rhie et al. 2021 Nature), and we re-estimated the divergence time across parrot phylogeny. In particular, the inclusion of kakapo allowed us to use a new fossil record (at the ancestor of kakapo and kea) for calibration, so we consider that the re-estimated time tree should be more accurate. This new time tree is also more similar to a recent study (Gelabert et al. 2020 Current Biology), and indicates that monk parakeet and blue-fronted amazon have a younger common ancestor (13.5 instead of 19.9 My). This further suggests chromosome changes in some lineage can be frequent.

Specific comments:

Line 88: Remove “R.”

R: Removed.

Line 91: An example of uncareful interpretation – neither of these three studies reveal this, but they are all in line with this suggestion. There are also several studies that are not in line with adaptive evolution of Y and they should also be highlighted.

R: We have revised this sentence into “Studies across many taxa with neo-XY chromosomes are in line with the suggestion that evolution of neo-Y chromosomes can play a role in resolving sexual antagonism and may affect speciation (Kitano et al. 2009; Zhou and Bachtrog 2012; Bracewell et al. 2017), though the neo-Y chromosomes often degenerate in the absence of recombination (Hough et al 2014)”.

Line 97: “recently”. This study is as recent as many of the studies in the sentence above.

R: “recently” deleted.

Line 100: What does “re-produce” mean?

R: We have added “by re-analyzing the Hi-C data”

Line 101: “reveals”. This is already known and well-described – use “confirm” instead.

R: Changed to “confirm”.

Line 111: PacBio or Nanopore? Give info in parenthesis.

R: Added “PacBio”

Line 114: Good!

R: Thanks for the positive remark.

Line 118-119: Even the latest zebra finch W (i.e. a species without any translocation) is longer than that (c. 20 Mb). Also, 13.8 Mb is surprisingly short as the W seems almost as long as the Z (which should be over 80 Mb) on the karyotypes (shown in this and other papers). What does this say about the quality of the W chromosome assembly presented in this paper? More details of the W chromosome are needed.

R: We have also realized that the size alone does not fully reflect the quality of the W chromosome assembly. The physical size of the monk parakeet W chromosome is very large, but given that the W chromosomes in the closely-related species are considerably small, we think the expansion of the monk parakeet W chromosome is a derived feature, resulting from the rapid expansion of satellite DNA (Furo et al. 2017). However, the error-prone long-reads still have a limited power to assemble young copies of satellite DNA sequences, so we were only able to assemble the repetitive sequences that are relatively older (exhibiting some divergence between copies). For instance, we estimated the copy number of the SatW20 to be 194,438 according to k-mer frequency of short-reads, but we failed to assemble them in the W chromosome. This is in part because the long-read assembler actually discards reads that have an extremely high copy number (because it will impose an extreme cost in pairwise alignment). During this revision, we emphasize more how many W-linked genes have been assembled, and added “harboring 92 protein-coding genes”.

Line 138. This is then not recent events - see comment above.

R: There are 4 fissions and 9 fusions occurring in the lineage leading to monk parakeet. Though we do not know the timing for each of the events, those 13 events all occurred within the last 19.9 Myr, after the split from blue-fronted amazon and monk parakeet. However, as the reviewer has pointed out, without sampling additional species, we should avoid considering the rate of karyotype changes as rapid.

Line 146-149: This result is presented in Figure 1a and is very difficult to understand as there is no explanation in the figure. Please expand this section here and provide understandable information in the figure (e.g. a y-axis with a label telling exactly what the vertical bars are showing). Also, are there also novel TEs in the other lineages presented? By only plotting parrot-specific TEs this illustration will look like this is something unique for parrots which may not be the case?

R: CR1 and LTR are two major TE groups in birds, taking up more than 90% of all bird TEs (also see supplementary fig. S4a). In Figure 1a, we show the activity of all CR1 and LTR subfamilies, but splitting CR1 into CR1-psi and CR1 excluding CR1-psi. This suggests (replying to the last question) that CR1-psi is quite abundant in parrots and has a recent expansion, and in other birds we don't see similar expansions of CR1 or LTRs. In this revision, we wrote in the legend: “The vertical bars show the timing of TEs (LTRs and CR1) insertions, with the height representing the frequency. CR1 is divided into CR1-psi and the other CR1. CR1-psi originated in parrots and continued to propagate in parrot lineages.”

Line 160: This is very important information for the idea the authors promote throughout the study. Thus, rather than just listing a lot of papers (many studies does not necessarily prove anything; do we even know whether they are independent?) explicit results from these studies should be given (what did they show, in what species, is it relevant to birds, etc.) – ideally in the main text as well as supplementary text.

R: We agree that we should not just list the papers, but should explicitly describe the major results of those studies. We now add in the results part:

“*ALC1* is a chromatin remodeler involved in DNA damage response (Ahel et al. 2009) and DNA end resection (Mejías-Navarro et al. 2020), and it has been demonstrated that deletion of *ALC1* impacts chromatin relaxation which is a crucial step in response to DNA damage (Sellou et al. 2016). *PARP3* responds to double-strand breaks (Beck et al. 2014; Belousova et al. 2018; Zarkovic et al. 2018), and has been shown to be involved in the repair of single-strand breaks in avian cells (Grundy et al. 2016). Studies show that the depletion of *PARP3* delayed the repairs of double-strand breaks (Boehler et al. 2011; Lindgren et al. 2013; Rodriguez-Vargas, Nguekeu-Zebaze, and Dantzer 2019) and exacerbated genome instability (Beck et al. 2019). Further studies demonstrated that *ALC1* collaborates with *PARP1*, another member of PARP (Poly(ADP-ribose) polymerase) family, on DNA repair (Tsuda et al. 2017; Juhász et al. 2020).”

Those studies came from different labs, suggesting *ALC1* and *PARP3* have been extensively studied. Moreover, many experiments involved gene knockouts (in mouse cell lines), providing direct evidence for the functions of *ALC1* and *PARP3*.

Line 165: What does “seems to be associated with” mean? Figure 1b clearly wishes to show this.

R: In Figure 1b we show multiple insertions of CR1-psi at the locus where *ALC1* was pseudogenized, however, we still do not have direct evidence that the gene loss was caused by the CR1-psi insertion. To clarify, we changed “seems to be associated with CR1-psi” to “coinciding with CP1-psi insertions”.

Line 173-174. This should/can be tested statistically using the position of the 72 genes.

R: Ideally we could identify genes located at the breakpoints of inversions throughout the genome which can allow us to do a Fisher’s exact test. However, since this is not the focus of this manuscript, we have not done such analyses, but we thank the reviewer for the good suggestion.

Line 189: Give explicit counts rather than inexact information such as “most”.

R: We added “32 out of 38”.

Line 191: Give time estimate in Myr rather than use information such as “recent”. Additional species (speciation event) need to be included in order to achieve better resolution on timing.

R: We added “occurred in less than 31.8 My”.

Line 212-213: This section needs better motivation. What is this hypothesis suggesting in terms of adaptive evolution and is there any previous study supporting it? It is not well-described how the insulation scores relate to Guerrero and Kirkpatrick’s hypothesis. What is an evolutionary breakpoint and how does it relate to fissions? Perhaps the section can be given as supplement or at least the figure?

R: We have extensively rewritten this section. We explained that fusions can be driven by natural selection because “the newly joined chromatins (could) have established frequent interactions due to new *cis*-regulation across the fusion sites”. This is one of the hypotheses raised by Guerrero and Kirkpatrick. We also added references where previous studies supported the hypothesis. Insulation scores are calculated using the Hi-C data, so not involved

in Guerrero and Kirkpatrick's hypothesis, but essentially it is useful to test whether interactions have been established across the fusion sites. In this revision, we used the term "fusion site" and "fission breakpoint" to refer to the position where two chromosomal segments join and one chromosome splits, respectively. We have also moved the figure to the supplements as the reviewer suggested.

Line 233. Again give time estimates rather than vague terms such as newly.

R: Here we do not intend to suggest how recent the event is, but only to indicate it is a derived feature.

Line 228-229: This is a very interesting result, novel to this study! But see comment above regarding Chr25.

R: We agree that the fusion of Gga25 to the sex chromosome is novel to this study. In Furo et al. 2020, it was observed that Gga25 binds to the largest chromosome in monk parakeet (during the revision we have repeated the experiment with new materials of wild individuals of monk parakeet). Unfortunately, Furo et al. misidentified the largest chromosome as Mmo1 which in fact should be MmoZ. We now explain this at line 359-361:

"It was previously observed that chr25 fused to a large macrochromosome (Furo et al. 2020), but the latter was wrongly identified as chr4 instead of chrZ".

Line 233: Check chromosome numbers and chromosome lengths. Gan et al. show that Chr1A is fused to Z.

R: The reviewer is right that in Gan et al. Chr1A was fused to the Z - we have modified the text accordingly.

Line 239, 245: This also suggests a relatively old age of this chromosome fusion.

R: We are aware that Gga11-GgaZ fusion was an ancient event (37 Myr ago at the ancestor of parrots), being one of the four ancient events we pointed out in line 301, and we did not intend to indicate in the manuscript that the Gga11-derived neo-sex chromosome is recent, but that the Gga25-GgaZ fusion should be much more recent (only in the lineage leading to monk parakeet).

Line 241: See comment above about Chr25.

R: We added "though it is possible chr25 only fused to the Z chromosomes". For chr11, it is certain that it has fused to both Z and W, because we have actually assembled the differentiated W-linked sequences.

Line 251-253. This is now revealed in this study, right? I believe you followed the previous definitions.

R: Correct, evolutionary strata are a well-defined concept in Neoaves sex chromosomes, but it was unknown how many parrot-specific strata there are (the three older strata were believed to be shared by all Neoaves because many Neoaves birds have been studied before (Zhou et al. 2014, Xu et al. 2019), not including parrots though). In this study, for the first time, we demarcated the parrot-specific strata (S3), and confirmed with phylogenetic evidence that the S3 evolved independently from that in passerines.

Line 257. The use of homologous ZW gene pairs here, suggest that the ZW chromosomes is not hemizygous as written above (line 239).

R: Though more than 83% of the neo-W genes have been deleted, a few genes (n=65) are still kept intact on the neo-W. To reflect this fact, we have replaced “hemizygous” in line 239 with “largely hemizygous”.

Line 260. Where do you show that this is a single stratum? I believe you need to evaluate the timing of recombination suppression along the fused region, before defining it as a single stratum.

R: In figure 4d we show the sequence similarity between the Z- and W-linked sequences. The Gga11-derived sequence is located at 85-103 Mb on the Z chromosome, and the entire sequence exhibits sequence similarity with the W homologs of around 86%. We use “sequence similarity” rather than the timing of recombination suppression to distinguish evolutionary strata, mainly because it is more straightforward to measure (sequence similarity or divergence should be correlated with divergent time). In this context, the Gga11-derived S3 is distinct from the other strata, for instance the S3 showing a sequence similarity of ~83% and S2 showing a sequence similarity of 79%. We now added “...exhibiting Z-W sequencing similarity of ~86%” in line 470-471.

Line 263. Do you show that Chr11 is attached to both Z and W?

R: In figure 4b we show Gga11 is present in both Z and W. Importantly, we have assembled Gga11 homologous (gametologous) sequenced on both the Z and W chromosomes. Note the individual we used for genome sequencing came from a local zoo in Fujian, China, while we used the wild individuals from the wild in Brazil for FISH experiments. This suggests the fusion of chr11 to Z and W is likely common to this species. For chr25, it is less certain as we ran out of female individuals for FISH experiments. To reflect this, we added “, though it is possible that chr25 only fused to the Z chromosome”

Line 267-268. The data in Fig. 4e is from a single gametolog at Chr11. This confirms that the fusion has this age and that recombination stopped at that particular position, but as mentioned just above this does not necessarily mean that this was the case for the whole region.

R: Please note that we used one representative gametolog (*CTCF*) to demonstrate the phylogenetic relationships of the gametologs, and it is not the case that we only relied on gametolog phylogenetics to infer the evolutionary history of strata. We in fact used the alignments of the entire Z and W chromosome to calculate the sequence divergence between them and demarcate the boundary of evolutionary strata. We believe the whole-chromosome data should be more informative because not many genes can be found on the W chromosome - so we used sequence divergence (including coding and non-coding regions) to define evolutionary strata. That being said, the phylogenetic relationships between the Z- and W-genes from multiple species are very useful to confirm the evolutionary strata we have defined. So we thought we could use one gene to demonstrate the phylogenetic relationships. During the revision, we examined all chr11-derived gametogs in monk parakeet, sun parakeet and kakapo, and collected those that were present in both monk parakeet and kakapo (sun parakeet genome

has a lower quality assembled with short reads, with many W-linked genes missing), and with their homologs present in at least three outgroup species. The 32 such gametologs are located across the neo-Z, and we produced phylogenetic trees and put them in a supplementary figure (supplementary fig. S13). In almost all gametologs, the W-linked genes cluster by parrot species.

Line 283. I do not follow “Additionally, W-linked gene loss leads to imbalanced dosage of the proto-sex-linked genes.” Additionally to what? Imbalance of what?

R: We revised this sentence to “The loss of W-linked gene loss also leads to imbalanced dosage of the proto-sex-linked genes relative to autosomes”. Hope now it’s clearer.

Line 286. “...fully...” Does this mean that your result suggest that S4 has evolved a mechanism to partly compensate? I think you should give data on mean \pm SD in different groups rather than using loose descriptions such as not fully.

R: If there was a complete lack of dosage compensation, the male-to-female expression ratio would be 2, and if there was a complete dosage compensation, the male-to-female expression ratio would be 1. In our data we noticed the male-to-female expression ratio ranges from 1.62 to 2.11 (2.11 is for testis/ovary ratios which shows large variance), indicating partial dosage compensation. We have added “with male-to-female expression ratios ranging from 1.58 to 2.02” in the main text, as well as adding a new supplementary table (supplementary table S5) showing all data.

Line 288, 292. Again, you have very little data on the rate of gene loss and dosage compensation (see comments above); thus, your study does not provide such a unique window into the temporal dynamics as now stated.

R: We have now provided gametologous data of all genes on the neo-sex chromosomes (supplementary figure S13), a table for dosage compensation results for both old and neo-sex chromosome (supplementary table S5), and accumulation of TEs on each evolutionary stratus (already presented in fig. 5d). Having assembled both the neo-Z and neo-W sequence allowed us to perform those analyses.

Line 297-316. How does this large size relate to you estimate of the W being 13.2 Mb (lines 117-118)? Line 302 – yes, repeat accumulation is also shown in several other avian W chromosomes. The discrepancy regarding the location of Chr25 between Furo et al. 2020 and this study needs to be clarified, before any further speculation. Also, even though the proportion of satellite repeats is higher on Chr25 in 2 of 3 species presented, there is also satellite repeats on all other chromosomes that could have acted as a source of satellite repeat expansion on the W.

R: We should point out that 13.2 Mb is the assembled size, but the actual size is probably much larger - unfortunately we do not have a reliable way to estimate the size. The fact that the W chromosome contains large arrays of satellite DNA, some of them are likely quite young, hampers the sequence assembly. Nevertheless, we identified and verified a 20-bp satellite DNA that can at least partially explain the sequence expansion of the W chromosome. It is true this is

not the only source, and we do not state that the 20-bp satellite DNA is the only reason for W chromosome expansion.

Line 323-324. I'm not sure why speciation is discussed here?

R: To avoid confusion we have removed this sentence.

Line 325. "...slightly..." What does slightly mean here? Is this data shown and tested?

R: We showed the data in line 142-143 and the supplementary figure S4. Because we only sampled two songbirds and five parrots, it is difficult to perform a statistical test. Here we show the data "9.6% vs. 7.8%" again in parentheses.

Line 326. "...speculate...". I like such careful wording regarding the conclusions in the discussion.

R: During the revision we try to rephrase sentences throughout the manuscript in a more careful way as suggested by the reviewer.

Line 343. TAD is not defined before. Also, it is not explained why this would be genetic drift rather than natural selection. Fission placements is a mechanism and not a process.

R: It is not straightforward to test whether it is natural selection that drives the fission. To avoid over-interpretation, we have removed this sentence.

Line 344. "One of the consequences...". There are cases of neo-sex chromosome formation in e.g. mammals and birds without additional inter-chromosomal rearrangements, so this is not a general requirement. Moreover, also in the present study it is premature to draw this conclusion as the fusion involving Chr11 is ancient within parrots and could have been the first to appear (i.e. not necessarily a consequence of frequent inter-chromosomal rearrangements (arguably, it is even impossible to exclude the alternative, i.e. that the fusion involving Chr11 was driving additional inter-chromosomal rearrangements)).

R: We fear that our uncaredful wording misled the reviewer. Here we only intended to refer to one specific consequence of fusions in parrots, that is, the fusion of Gga11 and ZW chromosome formed a pair of neo-sex chromosomes, instead of implying that inter-chromosomal rearrangements are necessarily responsible for neo-sex chromosome formation. To avoid misunderstandings, we have rewritten this sentence.

Line 363-523. In general, more information regarding the methods (and detailed results) would be desirable.

R: Thanks for the suggestions. In this revision, we have extensively re-organized and rewritten the methods section.

Figure 1a is beautiful but difficult to follow from the figure per se and the legends. A more informative figure description would be preferable.

R: We have added some descriptions of the figure in the legend.

Figure 2 is very nice, but some connections are difficult to follow. Details should be given as supplement.

R: We have uploaded files showing the pairwise alignments (in tabular format) to the Github repository which also hosts the codes we used in this study. (Note after zooming in one should be able to see all connections in figure 2)

Figure 3. The content of this figure is particularly difficult to condense (as is the result and its implication for chromosome evolution).

R: As the reviewer suggested above, we have moved this figure as a supplement.

Figure 4. Very nice and informative panel. However, gene trees (4e) should be given also for other gametologs along the chromosome. And as mentioned above the chr25-discrepancy to Furo et al. needs to be evaluated. 4f – do you have data explicitly showing that both 11 and 25 are physically linked to both Z and W (this does not need to be the case as e.g. Z1Z2W is also possible).

R: We have generated the gene tree of all S4 gametologs and put them into a supplementary figure (supplementary fig. S13). We have generated chromosome-level genome assembly of Monk parakeet which contains a single Z and a single W chromosome. As shown in 4a, both 11 and 25 were assembled into a single Z chromosome. On the assembled W chromosome, we have annotated gametologs that are homologous to genes on Gga11, suggesting the Gga11 has been physically added into the W chromosome. We have also responded to the similar comment listed above regarding chr25-discrepancy.

Figure 5a is very difficult to interpret (more information in the legend and possible in the figure would be desirable).

R: We have added a few more descriptions in the legend.

Figure 6. This figure shows a minor result (better as a supplement?).

R: We have moved this figure to the supplements.

Reviewers' Comments:

Reviewer #1:

Remarks to the Author:

I am satisfied that my points have been addressed and would gladly see a decision rest on whether the other reviewer's comments have been suitably addressed.

Reviewer #2:

Remarks to the Author:

The authors of NCOMMS-21-10399-T did a good job in replying to reviewer's comments but I still have couple of comments that need to be addressed:

About quality control, especially the short read data: You stated you didn't trim anything so I assume that quality of the reads was generally good when checked for example with FastQC? How about adapters? I hope those were at least removed? If this was done by the sequencing company, please mention that in the manuscript. Even if it is "just for expression" (but you also used the RNA-seq for annotation?) it is important that you use good quality data and know exactly what you are inputting in your bioinformatics pipelines so that reliable outcomes can be achieved. Same goes to raw reads downloaded from the NCBI. This also assures the reader that the best data was used in the study.

About summary tables: I would still add sequencing and mapping summary tables to the supplementary data in addition to the table S2. You can see examples for those e.g. from the Cooke et al 2017 Table S4, Weissensteiner et al. 2020 has also about the long reads. So information like how many reads were obtained from the sequencing company by sample, if they were mapped to genomes, what was the mapping % etc.

Lines 268-273: Which ones you generated and which ones were downloaded, please add citations. This info could also go to the methods, line 551 section.

Line 446 section: the figure S1 is fine but I would also briefly mention here which genomes were used for the Hi-C with citations.

Line 453: Which genome? Monk parakeet? It's good to remind the reader which species you are talking about as you have many bird species included. You could also add the species to the subheadings as well. For example "Hi-C scaffolding of monk parakeet draft genome", "Monk parakeet genome annotation", "Monk parakeet short read sequencing" etc.

Line 461: RNA-seq is mentioned the first time here so I would move the whole section about the RNA-seq next to the annotation. Now it confuses the reader.

Line 551, re-sequencing section: This needs more information. How these were sequenced?

Line 554: Which species?

Line 570: Which genomes?

Reviewer #3:

Remarks to the Author:

This paper uncovers substantial details, displayed by beautiful figures (e.g. figure 2 is great), of chromosomal patterns. Some of these patterns are already described in previous papers, including

that parrots have deviating and lower chromosome numbers than related species, and repeat expansions on the sex chromosomes. Still there are interesting novelties described such as which specific autosomes are fused (which was only partly known previously), that the whole of chr11 is fused to ZW, and the suggestion that chr25 is fused to ZW in one species. Furthermore, it is shown that lineage-specific repeats have accumulated on autosomes in general and on sex chromosomes in particular. The study further presents the results from gene expression of Z- and W-linked genes in males and females showing that the (now sex-linked) chr11-genes seem to evolve as genes located on the ancestral part of the sex chromosome (such a pattern has already been shown in publications of other avian neo-sex chromosome systems). That multiple autosomal fusions, and fusions to the sex chromosomes, occur are known from other bird lineages (Kretschmer et al. 2020 doi:10.3390/cells10010004 and Lo Cascio Sætre et al. 2021 doi:10.1093/gbe/evab212), so the phenomenon of recurrent fusions is not a novel discovery for birds, although it seems more commonly occurring in parrots.

Even after the revision, the paper suffers from circular arguments (statistical flaws), the main being that repeat expansions and gene losses that are defined as parrot-specific are implicated as drivers of parrot-specific chromosomal fusions (indeed, anything defined as lineage-specific will by necessity correlate to anything else also being lineage-specific). Nowhere in this study is cause and consequence understood; contrary to what is implicated, e.g. in the last line of Abstract. Also, few statistical tests are being presented – in fact it seems as p-values are preferentially given when significant (note however that the tests of ISs should probably be adjusted for multiple testing as 3 (or 4) species and tests are being made; lines 238-241), whereas test statistics and p-values are not provided for tests that probably would have rendered non-significant outcomes?

A main conclusion is that the fusions have resulted from neutral [rather than selective] processes (e.g. last line in the Abstract). This conclusion has low support because many (most?) of these fusions happened many million years ago and it is likely that the processes that have shaped the current chromosomal patterns (sequence diversity, gene expression and repeat landscape), i.e. the data being analysed here, are not (necessarily) the same processes causing the fusions. A comparison to a previous *Drosophila* study is made, but in that study the result (i.e., sex-specific evolution of sex-linked genes) supported the hypothesis making it a much stronger case. Overall, it is difficult to make conclusions about processes from genomic patterns as being done in the present study because many potentially processes may co-occur, and alternate spatially and temporally, throughout history of a lineage. If the authors still wish to make statements about evolutionary processes, they need to carefully consider avoiding misinterpretations, e.g. in this case by pointing out that historical processes including selection may be missed with the current data and therefore that either neutral or selective processes may have caused any of the patterns described. Sweeping statements (take home messages) as in the end of the Abstract should be avoided.

A question arises of how early strata were defined; based on sequence divergence patterns (figure 3d) these strata do not seem to be clearly separated. This can be quite important as this categorisation is used in downstream analyses (e.g. figure 4d). There is some cherry picking among results; e.g. showing results of a few selected gene trees only in the main text (figure 3e; but all gene trees are given as supplement), and focussing on few (two) of the many (74) lost genes (figure 1a-c). Based on the description of these two lost genes' functions in other species, understanding their potential role for chromosome alterations in birds should have been prioritized (an independent test could be to evaluate whether these two genes are lost also in the other bird species with multiple chromosome fusions and such data may be available; references above). Figure 4 still needs more explanation to be understood (e.g. in 4a, rows but not columns are explained; what are the lines in 4b; what is the unit on y-axis in 4d).

Reviewer #1 (Remarks to the Author):

I am satisfied that my points have been addressed and would gladly see a decision rest on whether the other reviewer's comments have been suitably addressed.

Reviewer #2 (Remarks to the Author):

The authors of NCOMMS-21-10399-T did a good job in replying to reviewer's comments but I still have couple of comments that need to be addressed:

About quality control, especially the short read data: You stated you didn't trim anything so I assume that quality of the reads was generally good when checked for example with FastQC? How about adapters? I hope those were at least removed? If this was done by the sequencing company, please mention that in the manuscript. Even if it is "just for expression" (but you also used the RNA-seq for annotation?) it is important that you use good quality data and know exactly what you are inputting in your bioinformatics pipelines so that reliable outcomes can be achieved. Same goes to raw reads downloaded from the NCBI. This also assures the reader that the best data was used in the study.

R: Yes, the company (Annoroad Gene Technology Co. Ltd) has trimmed low-quality bases and adapters, and removed low-quality read pairs. Indeed, when they delivered the "clean reads", they also attached a quality-check report (similar to the FastQC figures), for each sequencing library. To make it clear to readers, we added "Trimming for low-quality bases and adapters was also completed by Annoroad Gene Technology Co. Ltd, according to the filtering pipeline described in (Jiang et al. 2019)." at line 587-590. The filtering pipeline was developed by the company, and the details can be found in the reference (Jiang et al. 2019).

We have been fully aware of the pipeline and the required input data. To assemble the transcriptome for the purpose of gene annotation, we used the genome-guided (not *de novo*) assembly pipeline (HISAT2+StringTie), and "By default StringTie adjusts the predicted transcript's start and/or stop coordinates based on sudden drops in coverage of the assembled transcript." (from StringTie manual), and we used stringent parameters (-a 12 -j 5 -c 10) for transcriptome assembly.

For re-sequencing reads we downloaded from NCBI, we now explained "We did not trim these reads since they were used only to calculate sequencing coverage." at line 551.

About summary tables: I would still add sequencing and mapping summary tables to the supplementary data in addition to the table S2. You can see examples for those e.g. from the Cooke et al 2017 Table S4, Weissensteiner et al. 2020 has also about the long reads. So information like how many reads were obtained from the sequencing company by sample, if they were mapped to genomes, what was the mapping % etc.

R: we have updated the table S2 by adding the number of reads and the mapping (%) for each long- or short-read dataset.

Lines 268-273: Which ones you generated and which ones were downloaded, please add citations. This info could also go to the methods, line 551 section.

R: we added "Raw read data from Carolina parakeet (Gelabert et al. 2020), Hispaniolan amazon (Kolchanova et al. 2019), Puerto Rican amazon (Kolchanova et al. 2019) were downloaded from NCBI. We did not trim these reads since they were used only to calculate sequencing coverage." at line 551 section.

Line 446 section: the figure S1 is fine but I would also briefly mention here which genomes were used for the Hi-C with citations.

R: we added "to do scaffolding based on the draft genome (Ganapathy et al. 2014)".

Line 453: Which genome? Monk parakeet? It's good to remind the reader which species you are talking about as you have many bird species included. You could also add the species to the subheadings as well. For example "Hi-C scaffolding of monk parakeet draft genome", "Monk parakeet genome annotation", "Monk parakeet short read sequencing" etc.

R: we have changed "Genome annotation" to "Monk parakeet genome annotation", and changed "Short-read sequencing" to "Monk parakeet short-read sequencing", and changed "long-read sequencing" to "Monk parakeet long-read sequencing".

Line 461: RNA-seq is mentioned the first time here so I would move the whole section about the RNA-seq next to the annotation. Now it confuses the reader.

R: to make it clearer, we mentioned in the first section of the Method (**Sample collections**): "...and additional three males and females for RNA-seq. The RNA-seq data was used for genome annotations as well as quantifying the expression of sex-linked genes".

Line 551, re-sequencing section: This needs more information. How these were sequenced?

R: We added: "The re-sequencing data was generated in the same way as described in the section "Monk parakeet short-read sequencing".".

Line 554: Which species?

R: we added "of monk parakeet"

Line 570: Which genomes?

R: we added "monk parakeet".

Reviewer #3 (Remarks to the Author):

This paper uncovers substantial details, displayed by beautiful figures (e.g. figure 2 is great), of chromosomal patterns. Some of these patterns are already described in previous papers, including that parrots have deviating and lower chromosome numbers than related species, and repeat expansions on the sex chromosomes. Still there are interesting novelties described such as which specific autosomes are fused (which was only partly known previously), that the whole

of chr11 is fused to ZW, and the suggestion that chr25 is fused to ZW in one species. Furthermore, it is shown that lineage-specific repeats have accumulated on autosomes in general and on sex chromosomes in particular. The study further presents the results from gene expression of Z- and W-linked genes in males and females showing that the (now sex-linked) chr11-genes seem to evolve as genes located on the ancestral part of the sex chromosome (such a pattern has already been shown in publications of other avian neo-sex chromosome systems). That multiple autosomal fusions, and fusions to the sex chromosomes, occur are known from other bird lineages (Kretschmer et al. 2020 doi:10.3390/cells10010004 and Lo Cascio Sætre et al. 2021 doi:10.1093/gbe/evab212), so the phenomenon of recurrent fusions is not a novel discovery for birds, although it seems more commonly occurring in parrots.

Even after the revision, the paper suffers from circular arguments (statistical flaws), the main being that repeat expansions and gene losses that are defined as parrot-specific are implicated as drivers of parrot-specific chromosomal fusions (indeed, anything defined as lineage-specific will by necessity correlate to anything else also being lineage-specific). Nowhere in this study is cause and consequence understood; contrary to what is implicated, e.g. in the last line of Abstract. Also, few statistical tests are being presented – in fact it seems as p-values are preferentially given when significant (note however that the tests of ISs should probably be adjusted for multiple testing as 3 (or 4) species and tests are being made; lines 238-241), whereas test statistics and p-values are not provided for tests that probably would have rendered non-significant outcomes?

R: after revising the last sentence of the Abstract (now “Together, the combination of our genomic and cytogenetic analyses characterizes the complex evolutionary history of chromosomal rearrangements and sex chromosome in parrots”) and removing the last speculative sentence of the Discussions, we believe our current manuscript no longer suggests or implies any causal role of repeats or gene losses. We further added in the Discussions “Moreover, it remains elusive whether TEs play a causal role in promoting gene loss or chromosomal rearrangements, and we cannot rule out the possibility that TE expansion simply correlates with other evolutionary events that are specific to parrots.”. We hope our revised manuscript is now not leading the readers to the conclusions about the causal role of repeats.

In the figure S9, we use asterisks to denote significant results - for species without asterisks, the tests were non-significant. Now we made it clearer by adding “not significant in the absence of *” in the legend.

A main conclusion is that the fusions have resulted from neutral [rather than selective] processes (e.g. last line in the Abstract). This conclusion has low support because many (most?) of these fusions happened many million years ago and it is likely that the processes that have shaped the current chromosomal patterns (sequence diversity, gene expression and repeat landscape), i.e. the data being analysed here, are not (necessarily) the same processes causing the fusions. A comparison to a previous Drosophila study is made, but in that study the result (i.e., sex-specific evolution of sex-linked genes) supported the hypothesis making it a much stronger case. Overall, it is difficult to make conclusions about processes from genomic patterns as being done in the present study because many potentially processes may co-occur, and

alternate spatially and temporally, throughout history of a lineage. If the authors still wish to make statements about evolutionary processes, they need to carefully consider avoiding misinterpretations, e.g. in this case by pointing out that historical processes including selection may be missed with the current data and therefore that either neutral or selective processes may have caused any of the patterns described. Sweeping statements (take home messages) as in the end of the Abstract should be avoided.

R: we agree that a speculative statement should not be placed at the end of the abstract. Now we have changed the last sentence of the abstract to “Together, the combination of our genomic and cytogenetic analyses characterizes the complex evolutionary history of chromosomal rearrangements and sex chromosome in parrots”. We still leave some discussions about the role of TEs in the Discussions, but are refrained to make any conclusions without genetic evidence. For instance, in the last sentence of the first paragraph of Discussion, we wrote “Moreover, it remains elusive whether TEs play a causal role in promoting gene loss or chromosomal rearrangements, and we cannot rule out the possibility that TE expansion simply correlates with other evolutionary events that are specific to parrots.”.

A question arises of how early strata were defined; based on sequence divergence patterns (figure 3d) these strata do not seem to be clearly separated. This can be quite important as this categorisation is used in downstream analyses (e.g. figure 4d). There is some cherry picking among results; e.g. showing results of a few selected gene trees only in the main text (figure 3e; but all gene trees are given as supplement), and focussing on few (two) of the many (74) lost genes (figure 1a-c). Based on the description of these two lost genes' functions in other species, understanding their potential role for chromosome alterations in birds should have been prioritized (an independent test could be to evaluate whether these two genes are lost also in the other bird species with multiple chromosome fusions and such data may be available; references above). Figure 4 still needs more explanation to be understood (e.g. in 4a, rows but not columns are explained; what are the lines in 4b; what is the unit on y-axis in 4d).

R: We thank again for the helpful comments from the reviewer. To clarify how early evolutionary strata were defined, we have re-organised the writing of this section: present what is known for bird evolutionary strata before we present our data, hoping it will be easier to understand our results. Because “all Neoaves (including parrots and songbirds) shared the first three evolutionary strata (S0-S2)”, what we really need is to “demarcate the boundary of S3”. We have also explained in the legend that “The S0 and S1 were defined based on the homology of Neoaves S0 and S1 (Xu et al. 2019).”.

Regarding gametolog phylogeny, we would like to remind the reviewer that there are only four gametologs in S3 (table S4) - fortunately however, our previous experiences have demonstrated that *C18orf25* contains many informative sites for phylogenetic inference (Xu 2019 NEE), thus a useful gametolog to determine the phylogenetic relationships across species. Similarly, for S4, not all gametolog gene pairs contain sufficient informative sites (thus low bootstrapping support), therefore we had to rely on a few informative gene pairs - *CTCF* is one such gene, among others (it's not possible to show all informative genes in the main figure), and selecting another informative gene from the figures S13 does not change our conclusion that S4 had a common origin in parrots.

Regarding the focus on the two genes - we believe the other 72 genes might be associated with parrot traits other than chromosomal rearrangements (so not the focus of this manuscript); according to our literature review, only these two genes (*PARP3* and *ALC1*) may be functionally relevant with chromosomal rearrangements. We have indeed investigated whether these two genes were also lost in other birds with drastic chromosomal rearrangements - but obtained negative results. We now add "However, both *ALC1* and *PARP3* remain intact in falcons who also exhibit drastic chromosome rearrangements (data not shown). This implies that other genetic factors may have convergently contributed to chromosomal rearrangements in falcons".

For Figure4, we have added in the legend "The top three panels annotate whether a gene is Z- or W-linked and in which tissue/sex it is expressed. The left panel annotates which stratum a gene belongs to" for 4a, and "The solid line represents the regression relation between the expression of the Z and W, while the dashed line indicates the situation of equal Z-W expression" for 4b. The unit of 4d y-axis is percentage - we have now added "(%)" in the label title.

Reviewers' Comments:

Reviewer #2:

Remarks to the Author:

I have no further comments for the authors. I am satisfied with the replies.

Reviewer #3:

Remarks to the Author:

Lines 38-39: After reading the second sentence in the abstract, one is embedded in the hope that a main aim of the current study is to uncover the "evolutionary processes and underlying genetic mechanism of chromosomal rearrangements in parrots". Drawing such conclusions would have required very different methods than genome sequencing/cytogenetics of a few parrot species. In general, the last sentence of the abstract, "...the combination of our genomic and cytogenetic analyses characterizes the complex evolutionary history of chromosomal rearrangements and sex chromosomes in parrots...", seems to summarise reasonable aims much better, i.e. to describe chromosomal patterns in some parrot species. Indeed, this would have been a very exciting (and realistic) aim!

Line 44-45: "...with known functions in the repair of double strand breaks and maintenance of genome stability...". It would be preferable to be more careful here as this has not, I believe, been shown in birds or at least not in parrots. As you write in your answers to the previous comments "according to our literature review, only these two genes (PARP3 and ALC1) may be functionally relevant with chromosomal rearrangements", which suggests that you are not quite sure about the relevance of these genes in the context of the present study. Such more careful wording also follows later in the MS: "attracted our attention as they are related to repair of DNA damage and maintenance of genome stability" (line 167-168).

Lines 123-125: "This female genome of monk parakeet also contains a W chromosome that is 13.8 Mb long, harbouring 92 protein-coding genes." This seems rather short considering that chr11 has fused to the original W. Please add information of expected size from your cytogenetic work and the size of e.g. the zebra finch W sequence.

Line 154: What does "tend to be clustered by parrot species" mean? These TEs are parrot specific right, or does "tend" mean that they are only partially parrot specific? Please clarify.

Line 161: "...in all five parrot genomes...". There are six species in figure 1. Please clarify.

Lines 182-183: Describe how your result "the gene loss of five of them, including ALC1, coinciding with CR1-psi insertions" relate to the aim given at line 179, i.e. to "...investigate the mechanism of gene loss, we...".

Line 185: What is the exact definition of "a remarkable abundance of partial copies of CR1-psi", and why is it relevant? Is it not possible that e.g. 1 or 2 CR1-psi repeats would have been enough? Please clarify.

Lines 185-186: "This reflects the nature of a mutational hotspot in this region that may have initially led to pseudogenization of ALC1, but...". Please explain how you exclude with your data that ALC1 was not first going through pseudogenization and that CR1-psi accumulated after the strength of purifying selection on and around ALC1 had become weaker?

Lines 191-192: "This suggests the potential role of chromosomal rearrangements (inversions) in gene loss...". I agree, which however further shows the problem of interpreting the potential role of these gene losses in causing rearrangements.

Lines 231-233: "We next asked whether the chromosomal fusions were driven by natural selection to create a new linkage of formerly unlinked chromatin (Guerrero and Kirkpatrick 2014), and whether fissions occurred at random positions." Please explain how the importance of natural selection will be analysed and, in the end of the section, if the result gave support for this hypothesis or not.

Line 237: It would have been preferable to indicate how many tests have been conducted and which tests were statistically supported (perhaps add a supplementary table?).

Lines 264-266: "In fact, for the chr25-derived neo-sex chromosome we were not able to assemble the Z- and W-linked sequences separately, likely due to little sequence divergence between them...". It would have been interesting to know whether the heterogametic females have a higher degree of heterozygous sites at chr25 compared to the homogametic males, as expected for young strata by Palmer et al. (2019, Molecular Ecology).

Line 281: Correct "tdivergence".

Lines 321-324: "The young age of S4 also provides a unique window into the temporal dynamics of TE accumulation on the non-recombination sex chromosome, and we demonstrate that LTRs rapidly accumulate on the young strata while CR1 accumulate slower, but over time CR1 gradually increase its proportion as indicated by CR1 densities on older strata (Fig. 4d)." A suggestion would be to remind the readers about the age of S4 here, so that "young" and "rapid" can be more easily understood. Also, add "-s" to accumulate and increase.

Line 327: "The size of the W chromosome in monk parakeet is unusually large". See comment above about assembly size.

Line 354: "...can only lead to an underestimation of the frequency of chromosomal changes." I believe this is not true if none of the unassembled microchromosomes were rearranged.

Lines 358-361: Here it is suggested that CR1-psi accumulation led to chromosomal rearrangement and loss of genes, but then (lines 364-366) it is suggested that the loss of two genes led to chromosomal rearrangements. Please clarify which order of events, if any, is supported by data.

Line 368: Please clarify how natural selection and genetic drift are being distinguished. Also, regarding "fixed" – do you know that all rearrangements are actually fixed within species?

Lines 381-385: This is a long sentence that would benefit from rewording. What does "In fact" refer to? What does "the rapid gene loss further imposes the challenge of..." intend to suggest (why is rapid important and how rapid does it need to be to impose a challenge)? Please clarify.

Lines 389-390: "The lack of recombination of the W chromosome made it ineffective in purging the deleterious mutations." This is an unexpected ending of the discussion as this hypothesis was suggested a long time ago.

Figure 1 and its legend:

a. The information regarding TE is very difficult to understand from the figure and the legend – it is referred to as "The vertical bars show the timing of TEs (LTRs and CR1) insertions, with the height representing the frequency." What is meant with the timing and how was this decided? What is "frequency" (there is no unit given)? Does TE frequency correlate to sequencing technique/year of sequencing? How do we interpret that the accumulation of TEs in e.g. the Kakapo seems to start long before the speciation of the Kakapo?

b. What is the consequence for the general interpretation of the results that the Kea has very low frequency of CR1-psi but high frequency of fusions ($2n=62$, as the budgerigar with high CR1-psi)? This

is not mentioned/discussed in the main text.

c. The number of gene gains (0) and gene losses (74) is in the text noted as calculated from 5 species, but there are 6 species in the phylogeny.

Figure 2 and its legend:

a. In the legend it is stated that the "...12 chromosomes that experienced recurrent chromosomal rearrangements are highlighted in colors". I don't think this is the case. I'm uncertain about the meaning of "recurrent" here? E.g. chr11 seems to have fused once in the history of parrots, i.e. it is not a recurrent rearrangement. In contrast, e.g. chr2 (not highlighted in colour) seems to have been rearranged in both Blue-fronted amazon (a fission) and in Monk parakeet (2 fissions and 2 fusions). I suggest that all parrot specific rearrangements are highlighted (i.e., the ones listed in Supplementary figure S8).

Figure 3 and its legend:

a. In the legend to a): "The FISH result for chicken chr25 probe is shown in the Supplementary Fig. S8." This should read Supplementary Fig. S10.

b. In the legend to e) it may be indicated that additional gene trees are given in Supplementary Fig. S13.

c. In figure f): if the fusion of chr25 to W is not yet verified this uncertainty should probably be indicated here.

Figure 4 and its legend:

a. In figure a) it would be preferable to sort the y-axis (from stratum 0-4).

b. The legend describes the scatterplot in b) as "Three W-linked gametolog have elevated expression in female tissues relative to the Z-linked homologs. The solid line represents the regression relation between the expression of the Z and W, while the dashed line indicates the situation of equal Z-W expression". A better explanation of what is actually shown would be preferable. Also, although I doubt it will affect the result, calculating statistics on all tissues simultaneously should be avoided due to pseudoreplication.

c. In d): What is TE content (%) referring to? % of total sequence length? % of all TEs? Also, "LTRs are more abundant in younger strata" seems incorrect as S4 does not have the highest proportion of LTRs. Furthermore, it is still uncertain how different W-segments were allocated to different strata (in particular S0 and S1 seem difficult to separate due to chromosome rearrangements)?

Suppl. Figure S9 and its legend:

a. "We compared the insulation scores at the evolutionary breakpoints (EBPs) of fissions between emu and parrot genomes that show significantly lower values compared to the genomic average." This can be misinterpreted. Do you mean that you show the results from a comparison of ISs between emu and parrot at EBPs? Also, on the Y axis, Emu insulation score (IS) is indicated.

b. What was the rationale of using data from Emu and not e.g. chicken or zebra finch?

c. The statistics are still poorly provided in the MS, e.g. here only the p-values are given and lacking are information regarding the kind of statistical test and the sample size. Correct throughout the MS.

Reviewer #3 (Remarks to the Author):

Lines 38-39: After reading the second sentence in the abstract, one is embedded in the hope that a main aim of the current study is to uncover the “evolutionary processes and underlying genetic mechanism of chromosomal rearrangements in parrots”. Drawing such conclusions would have required very different methods than genome sequencing/cytogenetics of a few parrot species. In general, the last sentence of the abstract, “...the combination of our genomic and cytogenetic analyses characterizes the complex evolutionary history of chromosomal rearrangements and sex chromosomes in parrots...”, seems to summarise reasonable aims much better, i.e. to describe chromosomal patterns in some parrot species. Indeed, this would have been a very exciting (and realistic) aim!

R: We thank the reviewer for pointing out this concern. We wrote about a knowledge gap in parrot chromosomal rearrangements in the second sentence, and we hope our study has contributed a little bit to increasing our understandings of the “evolutionary processes and underlying genetic mechanism of chromosomal rearrangements in parrots”. We are pleased that the reviewer described the aim of our study as “exciting”.

Line 44-45: “...with known functions in the repair of double strand breaks and maintenance of genome stability...”. It would be preferable to be more careful here as this has not, I believe, been shown in birds or at least not in parrots. As you write in your answers to the previous comments “according to our literature review, only these two genes (*PARP3* and *ALC1*) may be functionally relevant with chromosomal rearrangements”, which suggests that you are not quite sure about the relevance of these genes in the context of the present study. Such more careful wording also follows later in the MS: “attracted our attention as they are related to repair of DNA damage and maintenance of genome stability” (line 167-168).

R: Yes, it's true we did not have prior knowledge about the role of *PARP3* or *ALC1* in birds before we conducted this genomics study. However, it's clear that these two genes are well studied in many systems (mice, mouse cell lines, chicken cell lines, etc.), therefore they became a major focus in our investigation. We hope this study may promote future studies on *PARP3* and *ALC1* in birds, such as chicken and zebra finches. We have now removed “attracted our attention as they”.

Lines 123-125: “This female genome of monk parakeet also contains a W chromosome that is 13.8 Mb long, harbouring 92 protein-coding genes.” This seems rather short considering that chr11 has fused to the original W. Please add information of expected size from your cytogenetic work and the size of e.g. the zebra finch W sequence.

R: we have added: “Given the cytogenetically large size of the W chromosome (Furo et al. 2017), some heterochromatic sequences are likely missing in the assembly.”

Line 154: What does “tend to be clustered by parrot species” mean? These TEs are parrot specific right, or does “tend” mean that they are only partially parrot specific? Please clarify.

R: we have changed “clustered by parrot species” to “clustered by species but not by CR1-psi copies”, meaning most CR1-psi copies originated in the terminal branches leading to species, but not in their common ancestor (see supplementary fig. S5c).

Line 161: "...in all five parrot genomes...". There are six species in figure 1. Please clarify.

R: Yes, it should six - we have corrected it.

Lines 182-183: Describe how your result "the gene loss of five of them, including ALC1, coinciding with CR1-psi insertions" relate to the aim given at line 179, i.e. to "...investigate the mechanism of gene loss, we...".

R: we added "to detect genomics changes at the loci of parrot gene losses" after "examined the sequences harboring the homologous genes in non-parrot genomes and compared their synteny with parrot genomes". We hope the writing is now clearer.

Line 185: What is the exact definition of "a remarkable abundance of partial copies of CR1-psi", and why is it relevant? Is it not possible that e.g. 1 or 2 CR1-psi repeats would have been enough? Please clarify.

R: we changed "a remarkable abundance of partial copies" to "multiple copies", as the reviewer pointed out the 1-2 CR1-psi should be sufficient.

Lines 185-186: "This reflects the nature of a mutational hotspot in this region that may have initially led to pseudogenization of ALC1, but...". Please explain how you exclude with your data that ALC1 was not first going through pseudogenization and that CR1-psi accumulated after the strength of purifying selection on and around ALC1 had become weaker?

R: we appreciate this interesting question. However, we fear that we cannot be completely confident to exclude any alternative scenario. We have added "likely" before "reflects" to allow any alternative interpretation.

Lines 191-192: "This suggests the potential role of chromosomal rearrangements (inversions) in gene loss...". I agree, which however further shows the problem of interpreting the potential role of these gene losses in causing rearrangements.

R: Yes, "gene losses" is likely not the only cause to rearrangements. We hope we have made this clear.

Lines 231-233: "We next asked whether the chromosomal fusions were driven by natural selection to create a new linkage of formerly unlinked chromatin (Guerrero and Kirkpatrick 2014), and whether fissions occurred at random positions." Please explain how the importance of natural selection will be analysed and, in the end of the section, if the result gave support for this hypothesis or not.

R: It's true that natural selection is difficult to be directly tested. The results, however, are in line with weak purifying selection. And we added at the end of the section: "Together, our analyses suggest that chromosomal rearrangements in parrots were likely not driven by natural selection, but likely shaped by purifying selection"

Line 237: It would have been preferable to indicate how many tests have been conducted and which tests were statistically supported (perhaps add a supplementary table?).

R: we believe that the supplementary fig. S9 has shown the results for all tests, including those that were not significant (without asterisks).

Lines 264-266: “In fact, for the chr25-derived neo-sex chromosome we were not able to assemble the Z- and W-linked sequences separately, likely due to little sequence divergence between them...”. It would have been interesting to know whether the heterogametic females have a higher degree of heterozygous sites at chr25 compared to the homogametic males, as expected for young strata by Palmer et al. (2019, Molecular Ecology).

R: This would have been a good way to test whether chr25 derived neo-sex chromosome has differentiated in females. The caution of doing so is that the assembled sequence was likely a chimeric one with both Z- and W-linked sequences (if Z and W recently diverged), therefore not an ideal reference to all variants - it would lead to an increase of male variants that are likely artifacts when male reads were mapped to the W-linked sequences.

Line 281: Correct “divergence”.

R: corrected.

Lines 321-324: “The young age of S4 also provides a unique window into the temporal dynamics of TE accumulation on the non-recombination sex chromosome, and we demonstrate that LTRs rapidly accumulate on the young strata while CR1 accumulate slower, but over time CR1 gradually increase its proportion as indicated by CR1 densities on older strata (Fig. 4d).” A suggestion would be to remind the readers about the age of S4 here, so that “young” and “rapid” can be more easily understood. Also, add “-s” to accumulate and increase.

R: we have added “formed ~31.8 MY ago”, and have added “-s” to accumulate and increase.

Line 327: “The size of the W chromosome in monk parakeet is unusually large”. See comment above about assembly size.

R: we have addressed this at line 125 by adding “Given the cytogenetically large size of the W chromosome (Furo et al. 2017), some heterochromatic sequences are likely missing in the assembly.”

Line 354: “...can only lead to an underestimation of the frequency of chromosomal changes.” I believe this is not true if none of the unassembled microchromosomes were rearranged.

R: this is true that the unassembled microchromosomes may not have any rearrangements. We have rewritten this sentence from “Though some microchromosomes were missing in the genome assemblies of blue-fronted amazon and budgerigar, they represent a tiny proportion of the genomes, and can only lead to an underestimation of the frequency of chromosomal changes” to “Some microchromosomes were missing in the genome assemblies of blue-fronted amazon and budgerigar, so we may have underestimated the frequency of chromosomal changes if the microchromosomes also experienced any rearrangements.”

Lines 358-361: Here it is suggested that CR1-psi accumulation led to chromosomal rearrangement and loss of genes, but then (lines 364-366) it is suggested that the loss of two

genes led to chromosomal rearrangements. Please clarify which order of events, if any, is supported by data.

R: we fear our data may not directly support the suggestion that CR1-psi accumulation led to chromosomal rearrangements, and have accordingly deleted this suggestion; however, we believe it's reasonable to suggest that CR1-psi accumulation has likely contributed to losses of at least some genes (see the text in the Result section). And this does not contradict the suggestion that the loss of *PARP3* and *ALC1* may be involved in promoting chromosomal rearrangements.

Line 368: Please clarify how natural selection and genetic drift are being distinguished. Also, regarding "fixed" – do you know that all rearrangements are actually fixed within species?

R: if chromosomal rearrangements were favored by natural selection, it should likely not be mediated through gene losses or TE insertions, the patterns of which do not seem to be associated with adaptive evolution. Given that our data was sampled at an interspecific level (different parrot species), and that we sampled monk parakeets from both Brazil and China, it's likely the chromosomal rearrangements were fixed in parrot species.

Lines 381-385: This is a long sentence that would benefit from rewording. What does "In fact" refer to? What does "the rapid gene loss further imposes the challenge of..." intend to suggest (why is rapid important and how rapid does it need to be to impose a challenge)? Please clarify.

R: We have split this sentence into two. "In fact" refer to the failure to detect female-specific selection. "Rapid" is not necessarily required for gene losses to impose a deleterious effect, but we used it to describe the fact that gene losses on the W chromosome were fast. What we intended to suggest here was that "rapid gene losses" on the W chromosomes are likely to have deleterious effects on females since birds are known to respond slowly to dosage imbalance due to sex chromosome differentiation (Sun et al. 2019 Genome Research).

Lines 389-390: "The lack of recombination of the W chromosome made it ineffective in purging the deleterious mutations." This is an unexpected ending of the discussion as this hypothesis was suggested a long time ago.

R: This is true - we have removed this sentence.

Figure 1 and its legend:

- The information regarding TE is very difficult to understand from the figure and the legend – it is referred to as "The vertical bars show the timing of TEs (LTRs and CR1) insertions, with the height representing the frequency." What is meant with the timing and how was this decided? What is "frequency" (there is no unit given)? Does TE frequency correlate to sequencing technique/year of sequencing? How do we interpret that the accumulation of TEs in e.g. the Kakapo seems to start long before the speciation of the Kakapo?
- What is the consequence for the general interpretation of the results that the Kea has very low frequency of CR1-psi but high frequency of fusions ($2n=62$, as the budgerigar with high CR1-psi)? This is not mentioned/discussed in the main text.
- The number of gene gains (0) and gene losses (74) is in the text noted as calculated from 5 species, but there are 6 species in the phylogeny.

- R: a. we apologize for the insufficient information in the legend (due to space limitations) regarding how the timing was estimated. We have actually described the methods in the Methods part, and we added “(see details in Methods)” to direct readers to refer to more details in the Methods. Because we only cared about the relative frequency of TE insertions, we did not add a Y-axis. The frequency indicates the amount of TE insertions during a certain period, and does not correlate to sequencing techniques. We have also changed the sentence from “The vertical bars show the timing of TEs (LTRs and CR1) insertions, with the height representing the frequency” to “The vertical bars show the frequency of TEs (LTRs and CR1) insertions during the evolution of bird species”, hoping it is clearer.
- b. This is a very interesting question. We believe it's mainly because kea has the lowest quality of genome assembly (published in 2014 with low coverage of short reads), and some CR1-psi copies are likely missing.
- c. the number “6” is correct - we have corrected it in the text.

Figure 2 and its legend:

- a. In the legend it is stated that the “...12 chromosomes that experienced recurrent chromosomal rearrangements are highlighted in colors”. I don't think this is the case. I'm uncertain about the meaning of “recurrent” here? E.g. chr11 seems to have fused once in the history of parrots, i.e. it is not a recurrent rearrangement. In contrast, e.g. chr2 (not highlighted in colour) seems to have been rearranged in both Blue-fronted amazon (a fission) and in Monk parakeet (2 fissions and 2 fusions). I suggest that all parrot specific rearrangements are highlighted (i.e., the ones listed in Supplementary figure S8).

R: To clarify, chr11 was highlighted in color not because of recurrent chromosomal rearrangements but because of sex chromosome fusions. We apologize for the misunderstanding, and have corrected the text.

Figure 3 and its legend:

- a. In the legend to a): “The FISH result for chicken chr25 probe is shown in the Supplementary Fig. S8.” This should read Supplementary Fig. S10.
- b. In the legend to e) it may be indicated that additional gene trees are given in Supplementary Fig. S13.
- c. In figure f): if the fusion of chr25 to W is not yet verified this uncertainty should probably be indicated here.

R: a. we have corrected the figure number.

b. we added “Additional gene trees are given in the Supplementary Fig. S13”.

c. we added “Whether chr25 has been added to the chrW remains to be verified”.

Figure 4 and its legend:

- a. In figure a) it would be preferable to sort the y-axis (from stratum 0-4).
- b. The legend describes the scatterplot in b) as “Three W-linked gametolog have elevated expression in female tissues relative to the Z-linked homologs. The solid line represents the regression relation between the expression of the Z and W, while the dashed line indicates the situation of equal Z-W expression”. A better explanation of what is actually shown would be

preferable. Also, although I doubt it will affect the result, calculating statistics on all tissues simultaneously should be avoided due to pseudoreplication.

c. In d): What is TE content (%) referring to? % of total sequence length? % of all TEs? Also, “LTRs are more abundant in younger strata” seems incorrect as S4 does not have the highest proportion of LTRs. Furthermore, it is still uncertain how different W-segments were allocated to different strata (in particular S0 and S1 seem difficult to separate due to chromosome rearrangements)?

R: a. We thank the reviewer for this suggestion, but still believe that clustering genes by expression regardless of the strata they belong to may be a better way to show the general pattern of gene expression.

b. we have changed “Three W-linked gametolog have elevated expression in female tissues relative to the Z-linked homologs” to “The X- and Y-axis show the expression of Z- and W-linked gametologs, respectively”.

c. the percentage indicates the proportion of repeats over the entire sequence - I believe this is a general way to describe “TE content”. It’s true S4 does not have the highest level of LTRs (likely because S4 is relatively younger), but we do not think it alters the overall trend. We would like to point out again (as we have explained the legend in Fig. 3) that “The S0 and S1 were defined based on the homology of Neoaves S0 and S1 (Xu, Auer et al. 2019)”, but not based on Z-W similarity data alone.

Suppl. Figure S9 and its legend:

a. “We compared the insulation scores at the evolutionary breakpoints (EBPs) of fissions between emu and parrot genomes that show significantly lower values compared to the genomic average.” This can be misinterpreted. Do you mean that you show the results from a comparison of ISs between emu and parrot at EBPs? Also, on the Y axis, Emu insulation score (IS) is indicated.

b. What was the rationale of using data from Emu and not e.g. chicken or zebra finch?

c. The statistics are still poorly provided in the MS, e.g. here only the p-values are given and lacking are information regarding the kind of statistical test and the sample size. Correct throughout the MS.

R: a. Yes, that’s correct, emu serves as an outgroup in a pre-fission status. In the main text we explained “Similarly, the breakpoint of fissions tend to be located in regions with lower ISs in the pre-fission chromosomes”.

b. It has been recently shown that emu preserved the most of ancestral karyotype of birds, and has the least chromosomal rearrangements (Liu et al. 2021 Genome research).

c. We provided the kind of test, e.g. Wilcoxon test in the main text. Now we have also added the test name and the number of fissions that were used for testing in the legend.